# Causal contribution of optic flow signal in Macaque extrastriate visual cortex for roll perception

Wenhao Li[1,2,4], Jianyu Lu[1,2,4], Zikang Zhu[1] & Yong Gu [1,2,3] ✉

Optic flow is a powerful cue for inferring self-motion status which is critical for postural control, spatial orientation, locomotion and navigation. In primates, neurons in extrastriate visual cortex (MSTd) are predominantly modulated by high-order optic flow patterns (e.g., spiral), yet a functional link to direct perception is lacking. Here, we applied electrical microstimulation to selectively manipulate population of MSTd neurons while macaques discriminated direction of rotation around line-of-sight (roll) or direction of linear-translation (heading), two tasks which were orthogonal in 3D spiral coordinate using a four-alternative-forced-choice paradigm. Microstimulation frequently biased animal's roll perception towards coded labeled-lines of the artificial-stimulated neurons in either context with spiral or pure-rotation stimuli. Choice frequency was also altered between roll and translation flow-pattern. Our results provide direct causal-link evidence supporting that roll signals in MSTd, despite often mixed with translation signals, can be extracted by downstream areas for perception of rotation relative to gravity-vertical.

Accurate perception of self-motion is vital for our daily behavior. Self-motion generates a global visual flow field, known as "optic flow" on the retina, providing one of the most powerful cues for inferring self-motion status for humans, nonhuman primates, and many other animals including insects[1–6]. Optic flow cue is also applied in modern autonomous navigation including unmanned ground and aerial vehicles in complex and dynamic environments[7,8]. Complex optic flow patterns typically simulate self-motion including translation along forward/backward (surge), up/down (heave), left/right (sway), and rotation around the three orthogonal motion axes respectively (roll, yaw, pitch). Among them, translation motion has been indicated to relate to heading perception and control of locomotion[3,9–11]. Rotations around earth-vertical (yaw) or lateral axis (pitch) are also involved in heading when active rotation of eyes is accompanied at the same time during self-motion. In this case, retinal or extra-retinal mechanisms are hypothesized to disentangle distorted optic flow on the retina and recover true heading[3,9,12–22]. Analogously, head turns produce a similar problem, and non-visual cues such as vestibular, proprioceptive, or

motor command have been proposed for help[23,24]. In contrast, rotation along longitudinal axis, i.e., line-of-sight (roll), happens frequently during locomotion for both terrestrial and aerial animals in which compensatory torsional eye movements are typically induced due to tilted head or whole body relative to gravity[25,26]. Thus, roll perception has been implied to be critical in postural control[25,27,28], subjective visual verticality[25,29–32], object rotation perception[33], and heading perception[5,34].

In the brain, a number of cortical areas have been indicated to process information from optic flow including the dorsal portion of medial superior temporal area (MSTd), ventral intraparietal area (VIP), 7a, the superior temporal polysensory area (STP), the smooth pursuit areas of the frontal eye field (FEFsem), the cingulate sulcus visual area (CSv), the visual posterior sylvian area (VPS), and V6[3,35–52]. Among these, MSTd has received heaviest investigation because single neurons in MSTd typically possess large receptive fields (RFs), suitable for integration of information over a wide field. Indeed, majority of MSTd neurons are modulated by complex optic flow patterns with first or

[1]CAS Center for Excellence in Brain Science and Intelligence Technology, Institute of Neuroscience, Chinese Academy of Sciences, 200031 Shanghai, China. [2]University of Chinese Academy of Sciences, 100049 Beijing, China. [3]Shanghai Center for Brain Science and Brain-Inspired Intelligence Technology, Shanghai 201210, China. [4]These authors contributed equally: Wenhao Li, Jianyu Lu. ✉e-mail: guyong@ion.ac.cn

higher orders[33,53,54]. Many neurons prefer leftward and rightward laminar motion, which is ideal for discrimination of heading direction varied in fine steps around straight ahead[55]. Moreover, MSTd's neuronal activity is functionally coupled with animals' perceptual judgments about their experienced heading directions on a trial-by-trial basis[56–59]. Importantly, causal manipulation experiments have provided solid evidence showing that linear translation motion signals in MSTd causally contribute to heading perception[59–62]. In humans, lesions to the homologous area seriously impair navigation ability using optic flow[63]. The above-mentioned studies have mainly focused on the translation signals that have been implied in heading perception. In contrast, although earlier studies have reported rotation signals including roll in MSTd, more direct evidence supporting their functional implications in perception is basically lacking. In the current study, we aim to fill in this gap by designing a four-alternative forced choice (4-AFC) roll and translation direction discrimination task, in which macaques need to identify flow patterns as well as direction in roll (clockwise, CW versus counter-clockwise, CCW) and translation (leftward versus rightward). We apply electrical microstimulation to selectively manipulate activity of a population of clustered MSTd multiunits (MUs) and examine subsequent changes in the animals' behavioral performance. Our experimental paradigm allows us to underpin functional roles of roll signals in MSTd in self-motion perception simulated from optic flow.

## Results

### Behavioral task and performance

We trained two macaques to perform a combined roll and translation direction discrimination task with a 4-AFC paradigm (Fig. 1). The visual stimuli were selected from a 3-dimensional (3D) "spiral space" defined by two planes of roll and translation that are perpendicular to each other (Fig. 1a, blue and red, respectively). In each trial, the animals maintained central fixation, while optic flow stimuli over the whole visual display (90° × 90°) indicated about flow pattern (roll or

translation) and its motion direction (Fig. 1b). Specifically, the animals were required to make up or down saccade for roll direction discrimination task, and left or right saccade for linear translation discrimination task (Fig. 1c). Note that an ambiguous stimulus condition was provided without any information indicating correct flow pattern, as well as the motion direction (Fig. 1c, middle black point), and thus random reward was delivered in this condition. Otherwise, reward was delivered only when animals made correct judgments about motion direction of the given flow pattern. All stimulus conditions were interleaved across trials in one experimental session, yet in some cases, roll and translation tasks were performed in separate blocks with a classic two-alternative forced choice (2-AFC) paradigm.

For both roll and translation tasks, we also provided two different versions with respect to the way of task difficulty manipulation. In the "fine" version (Fig. 1d, left panel), an expanding optic flow pattern with focus of expansion (FOE) located at the center of the visual display was served as the reference. Difficulty in the roll discrimination task was manipulated by small rotary components ($\theta_R$) superimposed onto the reference of the expanding flow, leading to a high-order (second-order) of "spiral" flow-pattern. Difficulty in the translation discrimination task was manipulated by FOE varied along the horizontal meridian in fine steps around the reference. In contrast in the "coarse" version (Fig. 1d, right panel), motion direction was defined by a low-order (first-order) of pure CW or CCW roll, or pure leftward or rightward translation without any expansion component. In these cases, the reference was zero-coherence random-dot-motion (RDM) stimulus, and difficulty of the task was controlled by coherence varied between 0% and 8%.

The animals' behavioral performance was quantified by psychometric functions. We first assessed the animals' direction discriminability within each flow pattern by only including trials with correct choice for each respective flow pattern. This was shown as psychometric functions typically seen in classical 2-AFC tasks in Fig. 2a (color curves and black symbols), which clearly demonstrated that after

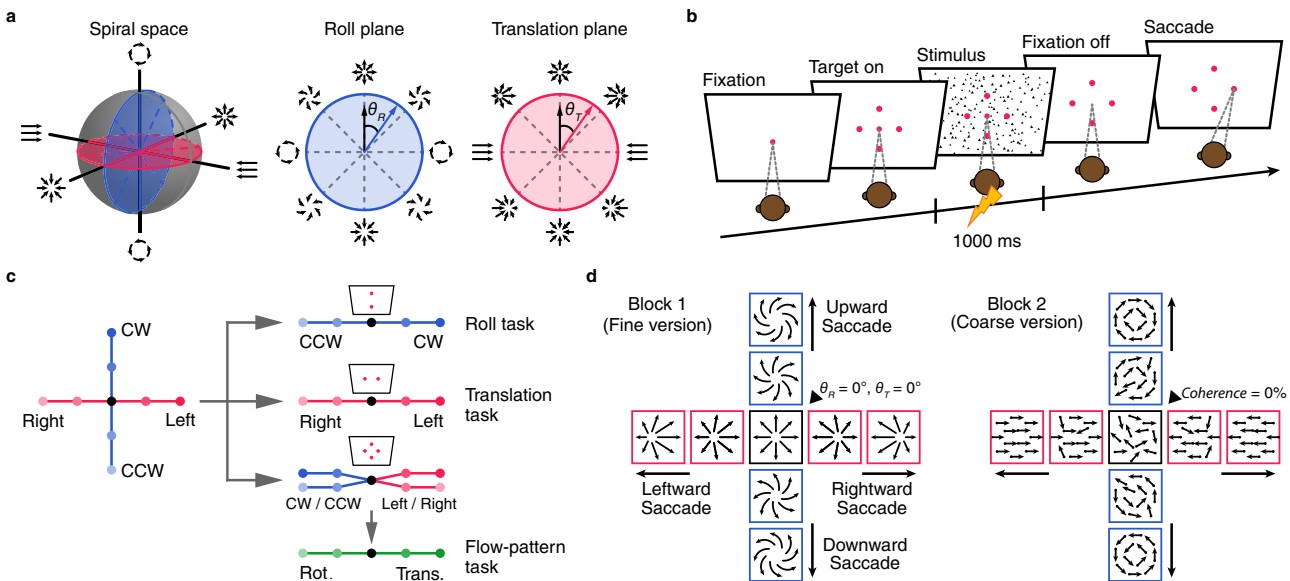

**Fig. 1 | Experimental protocol. a** Schematic plot of the 3D spiral space (left), roll plane (middle) and translation plane (right). Complex optic flow pattern used in tuning measurement and discrimination task were selected from the two perpendicular planes based on deviation in rotation component ($\theta_R$) and translation component ($\theta_T$) from straight forward ($\theta_R = 0°, \theta_T = 0°$). **b** Events flow in the main 4-AFC discrimination tasks. Each trial was initiated with a center fixation point and four choice targets appeared symmetrically around the fixation point (top, down, left, and right). **c** Each 4-AFC task can be divided

into roll direction discrimination task (CCW vs. CW), translation direction discrimination task (rightward vs. leftward), and flow-pattern discrimination task (roll vs. translation). **d** Fine (left) and coarse (right) version of the tasks. In the fine version, motion stimulus was expanding forward ($\theta_R = 0°, \theta_T = 0°$) with a small CCW ($-\theta_R$) or CW ($+\theta_R$), or a small rightward ($-\theta_T$) or leftward ($+\theta_T$) motion component. In the coarse task, only pure roll ($\theta_R = \pm 90°$) or laminar motion ($\theta_T = \pm 90°$) was presented, and the visual coherence was varied across trials.

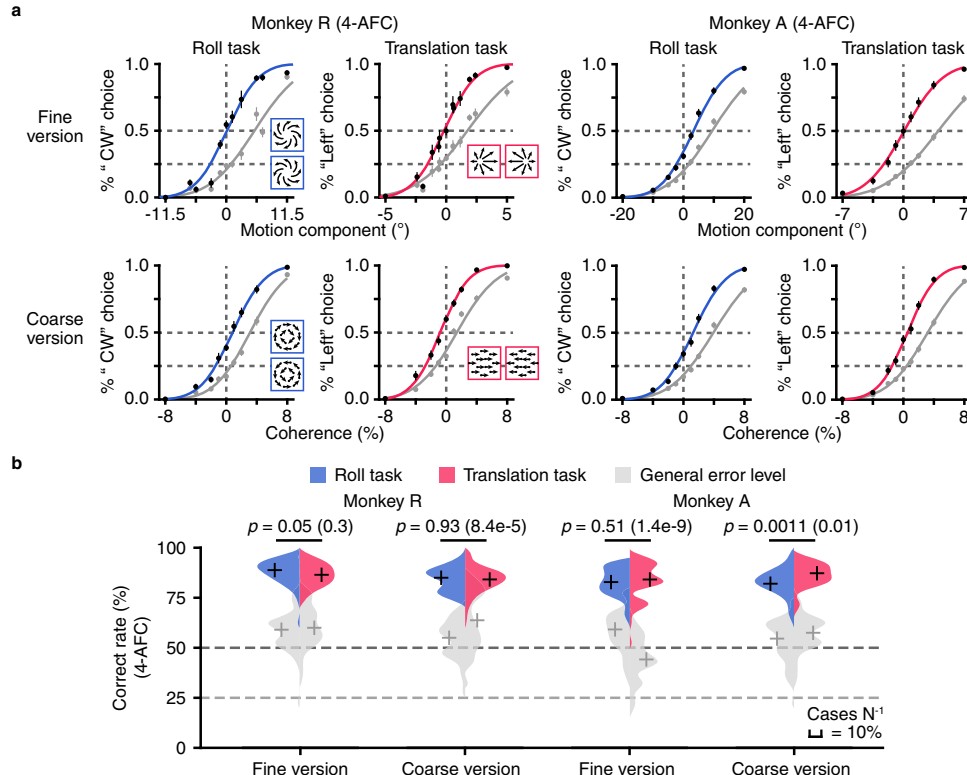

**Fig. 2 | Behavioral performance in 4-AFC task for monkey R and monkey A.**
**a** Average psychometric functions for roll task (blue) and translation task (red) under fine and coarse versions across animals. In roll task, psychometric functions were plotted from the proportion of "CW flow pattern" choice as a function of the "rotation axis" (rotation component $\theta_R$ in the fine version; coherence in the coarse version). In translation task, psychometric functions were plotted from the proportion of "Left flow pattern" choice as a function of the "translation axis" (translation component $\theta_T$ in the fine version; coherence in the coarse version). Color curves and black dots: only trials with correct flow pattern choice were included. Gray symbols: similar data format albeit including all trials in each flow pattern. For monkey R in fine-roll paradigm, $n = 68$, $\mu = 0.13°$, $\sigma = 4.7°$ ($\mu = 0.91°$, $\sigma = 6.6°$ for general error psychometric function, the same hereinafter); in coarse-roll paradigm, $n = 51$, $\mu = 0.82\%$, $\sigma = 3.3\%$ ($\mu = 0.61\%$, $\sigma = 3.7\%$); in fine-translation

paradigm, $n = 68$, $\mu = -0.17°$, $\sigma = 2.0°$ ($\mu = -0.47°$, $\sigma = 3.3°$); in coarse-translation paradigm, $n = 89$, $\mu = -0.78\%$, $\sigma = 3.0\%$ ($\mu = -1.4\%$, $\sigma = 4.1\%$). For monkey A in fine-roll paradigm, $n = 65$, $\mu = 3.3°$, $\sigma = 8.3°$ ($\mu = 1.8°$, $\sigma = 11.3°$); in coarse-roll paradigm, $n = 64$, $\mu = 1.3\%$, $\sigma = 3.4\%$ ($\mu = 1.0\%$, $\sigma = 4.4\%$); in fine-translation paradigm, $n = 65$, $\mu = 0.089°$, $\sigma = 3.5°$ ($\mu = 0.79°$, $\sigma = 4.8°$); in coarse-translation paradigm, $n = 64$, $\mu = 0.52\%$, $\sigma = 3.0\%$ ($\mu = 0.41\%$, $\sigma = 4.1\%$). Error bars, SEM. **b** Histograms show distributions of the correct rate (CR) for roll task (blue) and translation task (red) under fine and coarse versions across animals based on trials with correct flow pattern choice, and the general error level when all trials in each flow pattern were included (gray). Crosses, median CR; Horizontal short lines, $p$ value for two-tail Wilcoxon rank-sum test between roll task and translation task; Dark dashed line, 50% guess rate for roll and translation task; Light dashed line, 25% guess rate for general error level; Scale bar, 10% cases $N^{-1}$.

training, both monkeys discriminated directions well in either roll or translation task under fine and coarse versions. However, in 4-AFC task, the animals also made a significant number of wrong choices across flow patterns, particularly for the stimuli close to the ambiguous condition. To assess this type of error, we plotted another format of psychometric functions based on all trials within each task condition (gray curves and symbols in Fig. 2a). If the animals always choose correct flow pattern, we expect to see the two types of psychometric functions 100% overlapping. The clear shift of the two functions indicated that such inter-flow pattern error existed, particularly for more ambiguous stimuli conditions. Yet the magnitude appeared modest, indicating that overall the animals learned to correctly identify flow patterns, as well as the corresponding motion directions on a trial-by-trial basis. This was also reflected in the overall correct rate (CR), as computed based on either intra-flow pattern error (Fig. 2b, color shaded areas), or general error that included inter-flow pattern wrong choice (Fig. 2b, gray shaded areas), which was roughly equal across tasks (roll and translation), context (fine and coarse), and animals (monkey R and monkey A).

## Mixed roll-translation signals are predominant in MSTd

To guide microstimulation at proper sites, we first measured basic tuning properties in MSTd. Activities of 533 multiunit (MUs) from two

animals (Monkey R: $n = 444$, monkey A: $n = 89$) were recorded during simple fixation without requirement for any behavioral judgment (Supplementary Fig. 1a). Similar to previous studies, RFs of majority of the MUs were fairly large (mean diameter: ~52° for monkey R; ~54° for monkey A), frequently covering the fovea (Supplementary Fig. 1b, c) which was in sharp contrast to those in the neighboring middle temporal area (MT)[64–66] and the lateral part of MST (MSTl)[67–71] that typically contain much smaller and restricted RFs.

In Fig. 3a, the example MU showed clear modulation in both roll and translation planes ($p_{Roll} = 2.1e-5$, $p_{Translation} = 5.5e-9$, one-way ANOVA), leading to a resultant vector located somewhere in the 3D spiral space. Such a MU was defined into the "spiral" category[33,57,72]. Across population, we found a large proportion of MUs, from half to two thirds (monkey R: 54.5%; monkey A: 69.7%), contained both significant roll and translation signals, which were defined based a fairly strict criterion ($p_{Roll} < 0.001$, $p_{Translation} < 0.001$, one-way ANOVA, see Method). In contrast, MUs with only one of the modulation components occupied only a small proportion (Roll-only < 15%: $p_{Roll} < 0.001$, $p_{Translation} > 0.001$, one-way ANOVA; Translation-only <13%: $p_{Roll} > 0.001$, $p_{Translation} < 0.001$, Fig. 3b). Thus, MSTd is dominated by "spiral" MUs that contain both roll and translation signals. Moreover, roll signals were comparable to the translation signals in several aspects (Supplementary Fig. 1d, e)

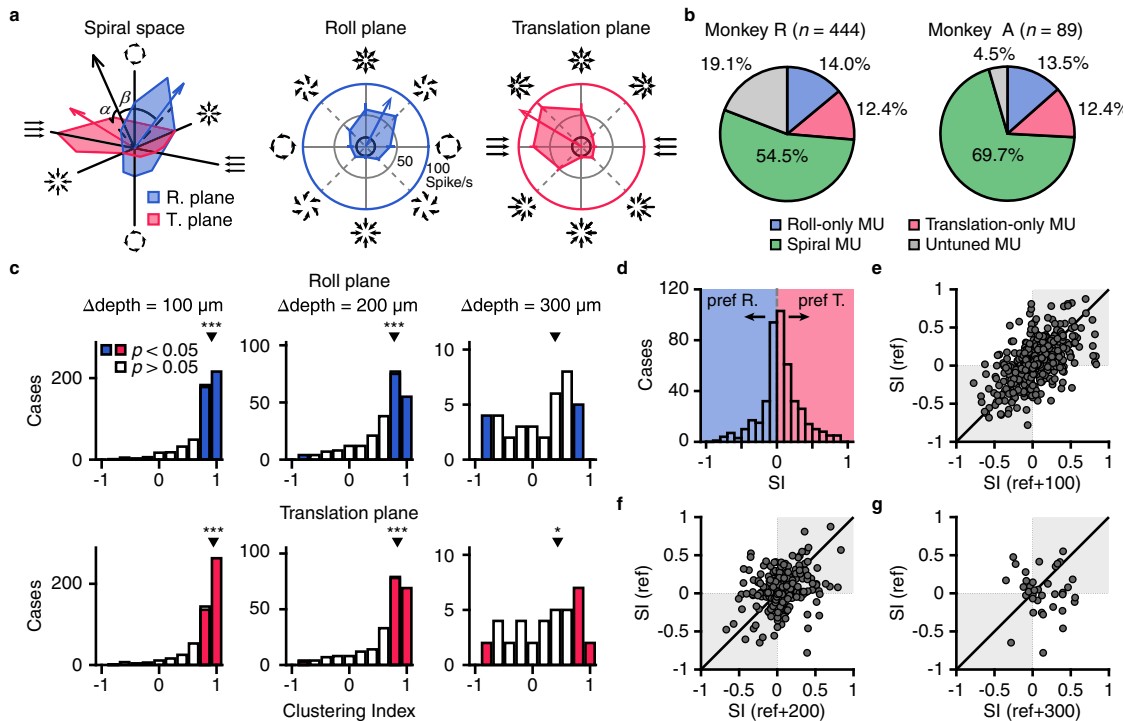

**Fig. 3 | Tuning properties of MSTd MUs. a** Polar plot of tuning for an example MU in the 3D spiral space (left), roll plane (middle) and translation plane (right). Based on vector sum of MU activity, the black arrow in the 3D spiral space, blue arrow in the roll plane and red arrow in the translation plane represent residual vector and indicate the preferred direction, respectively. $\alpha$ and $\beta$ in spiral space represent the angle between the residual vector and the roll plane and translation plane, respectively. Polar angle, the rotation component or translation component; Radius, firing rate; Thick black circles, spontaneous firing rate; R. plane, roll plane; T. plane, translation plane. **b** Proportion of different cell types of MSTd for monkey R and monkey A. **c** Distributions of clustering index of roll (up) and translation (bottom)

tuning across sites ($\Delta$depth = 100 μm, 200 μm, 300 μm) along an electrode penetration. Filled bars, clustering index significantly different from zero ($p < 0.05$, Pearson's correlation). Inverted triangle, median; Asterisk, median significantly different from zero (*$p < 0.05$; ***$p < 0.001$, two-tail sign test). In roll plane: $p = 1.0e$-92, 1.0e-30 and 0.19 for $\Delta$depth = 100 μm, 200 μm and 300 μm, respectively. In translation plane: $p = 7.4e$-96, 4.8e-32 and 0.047 for $\Delta$depth = 100 μm, 200 μm and 300 μm, respectively. **d** Distribution of spiral index. **e**–**g** Scatter plot of spiral index for nearby sites of the same MU separated by 100 μm (Pearson's correlation coefficient = 0.55, $p = 3.3e$-43), 200 μm (Pearson's correlation coefficient = 0.29, $p = 5.8e$-6), and 300 μm (Pearson's correlation coefficient = −0.026, $p = 0.88$).

including maximum firing rate, variance-to-mean ratio (Fanor factor), and direction discriminability (roll: CW verse CCW; translation: leftward verse rightward) as defined either by $d'$ or ROC analysis.

Microstimulation experiments typically require that sites with similar tuning properties are clustered within a local range in order to artificially evoke a consistent signal to affect the animals' decision[59,61,62,65,73–77]. Previous studies have demonstrated that the translation, or "heading" signals are clustered well in MSTd[59,61,62,74], yet it is unknown how rotation, or spiral signals are clustered. To assess this, we used two methods. In the first method, we measured tuning similarities by computing the Pearson's correlation coefficient for tuning curves at adjacent sites in individual planes of roll and translation. We found similar to translation tuning, rotation tuning was also clustered well within a local range (≤200 μm, Fig. 3c). In the second method, we assessed spiral tuning by computing a spiral index for each MU site (see Method). Briefly, a residual vector was first computed by vector sum for neural activities in the 3D space (Fig. 3a, black arrow). Tuning curve in at least one of the planes (roll or translation) needs to be significantly modulated as assessed by one-way ANOVA ($p < 0.05$). A spiral index was then defined based on comparison of the angles of the residual vector from each plane (see Eq. 1, Method). Spiral index ranged between [-1 1], with 1 indicating strong translation preference and -1 indicating strong roll preference. If the spiral index of a MU is around 0, it implies that the MU responds to both roll and translation motion equally well. We found that the distribution of the values peaked around zero (Fig. 3d), indicating that spiral MUs that contained both roll and translation signals were dominant in MSTd. Importantly, spiral index was similar for nearby sites (≤200 μm, Fig. 3e, f), again

illustrating that neural signals modulated by complex optic flow patterns including both translation and rotation components are clustered in MSTd. This makes MSTd an ideal place for applying microstimulation experiments as illustrated in the following sections.

**Microstimulation biased roll direction judgment**

To examine whether and how roll signals in MSTd contribute to the animals' perception of rotation around line-of-sight, a weak current (20 or 40 μA, see Method) was delivered to activate a small population of units on half of trials that were randomly interleaved with the rest without microstimulation (control trials). Figure 4a showed microstimulation effects on the animal's perceptual judgment in the 4-AFC paradigm when stimulating the same example site as shown in Fig. 3a. The left and right columns represent the microstimulation effects as a function of natural behavior and preferred stimulus, respectively. This site preferred CW roll rotation, and microstimulation indeed biased the animal's perceptual judgment towards "CW" choice (Fig. 4a, up panel, left column), that is, the preferred direction (PD) choice (Fig. 4a, up panel, right column). As a comparison, the same site preferred rightward moving flow, and in the translation direction discrimination task, microstimulation indeed biased the animal's perceptual judgment towards rightward-motion choice (Fig. 4a, down panel). Such an effect was similar for both types of psychometric functions constructed based either on trials with correct flow pattern choice (solid curves and filled symbols), or on all trials within each flow pattern (dashed curves and cross symbols). Thus for this example site, both roll and translation signals were read out by downstream areas in a way concordant with their preferred directions in each individual task. For

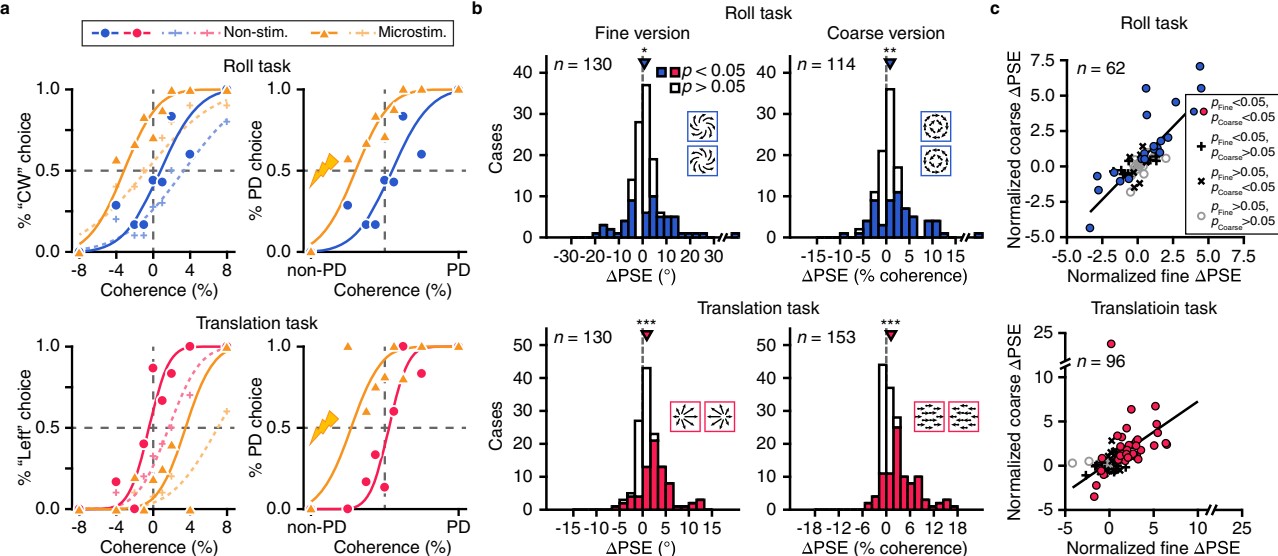

**Fig. 4 | Microstimulation effect on PSE shift in 4-AFC roll and translation task.**
**a** Example psychometric function for non-stimulated trials (control trials, blue symbols for roll task and red symbols for translation task) and stimulated trials (orange symbols) from the same site as shown in Fig. 3a. Solid curves and filled dots, psychometric function including trials only with correct flow pattern choice. Dashed curves and cross symbols, general error psychometric function including all trials within each flow pattern. Left column: microstimulation effect as a function of the actual stimulus as in the behavioral performance (Fig. 2a). Right Column: microstimulation effect as a function of preferred direction (PD) of the stimulated site. **b** Summary of microstimulation-induced PSE shifts (ΔPSE) in 4-AFC task for fine and coarse versions. Positive ΔPSE value represents shifts in the expected direction and negative value represents shifts in the opposite direction. Filled and open bars represented significant ($p < 0.05$, probit regression) and nonsignificant ($p > 0.05$, probit regression) ΔPSE, respectively. Inverted triangles, median ΔPSE; Asterisks, median ΔPSE significantly different from zero (*$p < 0.05$; **$p < 0.01$; ***$p < 0.001$, two-tail sign test). Fine-roll, $p = 0.044$; coarse-roll, $p = 1.0e\text{-}3$; fine-translation, $p = 5.8e\text{-}7$; coarse-translation, $p = 2.6e\text{-}5$. Dashed vertical lines: zero ΔPSE. **c** Comparison of normalized ΔPSE under fine and coarse versions. Filled circles: sites with significant ΔPSE under both versions; Crosses: sites with significant ΔPSE under either one version; Open circles: sites without significant ΔPSE in either version; Black lines: linear regress fit.

this "expected" type of case, we assigned a positive value for the induced point of subjective equality (PSE) shift (i.e., +ΔPSE), and assigned a negative value for the other "unexpected" cases (−ΔPSE) when the induced PSE shift was in the direction opposite to the preferred direction of the activated sites. For simplicity, the following analyses within this section would be only applied on psychometric functions with correct flow pattern choice. Results were similar when using general error that included wrong inter-flow pattern choice (Supplementary Fig. 2). The latter type of error would be specifically analyzed in an independent section.

The example in Fig. 4a represents the overall trend of microstimulation effects in our population data (Fig. 4b). Because the two animals showed similar results (Supplementary Fig. 3a, b), data from the two animals were pooled together to give more statistical power. For the same reason, data were also pooled from two current amplitude conditions (see Method, and Supplementary Fig. 4a, b). The finalized 4-AFC task dataset includes 130 sessions in the fine-roll paradigm, 114 sessions in the coarse-roll paradigm, 130 sessions in the fine-translation paradigm, and 153 sessions in the coarse-translation paradigm.

In the roll task with fine version, microstimulation frequently induced PSE shift, with a statistically significant effect ($p < 0.05$, probit regression) in nearly half of the cases ($n = 56$, 43.1%, Fig. 4b, top left panel). Among these, nearly two thirds (36/56, 64.3%) were in the expected direction with respect to the tuning preference of the stimulated sites. On average, the median PSE shift across all sites was significantly greater than 0 (median ΔPSE = 0.75°, $p = 0.044$, two-tail sign test). Such a trend was similar in the coarse version (Fig. 4b, top right panel). In particular, microstimulation induced significant PSE shift in 60 cases (52.6%), and more than two thirds (42/60, 70.0%) were in the expected direction. On average, the overall PSE shift was significantly larger than 0 (median ΔPSE = 0.82%, $p = 0.001$, two-tail sign test). On 62 cases, we have run both fine and coarse versions. This dataset allows us to compare microstimulation effects between the

two task versions directly on a site-by-site basis. We found that induced PSE shift was highly correlated and indistinguishable from each other (Fig. 4c, top panel, $p = 4.1e\text{-}15$, Pearson's correlation), suggesting that fine or coarse task version does not cause difference in contribution of the rotation signals in MSTd to roll perception.

As a comparison, microstimulation also induced significant PSE shift in the translation task under both fine and coarse versions (Fig. 4b, c, bottom panels). The overall shift was also in the expected direction (fine: median ΔPSE = 0.91°, $p = 5.8e\text{-}7$; coarse: median ΔPSE = 1.2%, $p = 2.6e\text{-}5$, two-tail sign test), with an effective size relatively larger than that in the roll task. This was reflected in the scatter plot when directly comparing microstimulation-induced PSE shift between roll and translation tasks (Supplementary Fig. 3c).

We then further examine what factors may have affected microstimulation effects. Intuitively, neurons do not contribute to perceptual decision in a uniform way, but accord to their sensitivity that is relevant to the task. For example, previous studies have indicated that neurons preferring leftward or rightward laminar motion are more sensitive to directions varied around a straight forward or upward reference[55,78]. Analogously, it is also likely that neurons with preference closer to CW or CCW may be more functionally linked to roll perception. To test this hypothesis, we examined the dependence of induced PSE shift on the tuning preference of the MUs being activated by microstimulation. In the roll task, we found that stimulating sites with preference to pure rotation (i.e., $\theta_R = \pm 90°$) indeed tended to evoke PSE shift in the expected direction, whereas stimulating sites with preference close to the reference was more likely to evoke PSE shift in the unexpected direction (Fig. 5a, b, left panels). Similarly, this trend also existed in the translation task (Fig. 5a, b, right panels). These results were similar in the fine and coarse versions in both animals (Supplementary Fig. 5). Note that proportion of MUs preferring CW and CCW was a bit unequal, which was probably due to a sampling bias, particularly in one of our animals (monkey A, Supplementary Fig. 1f).

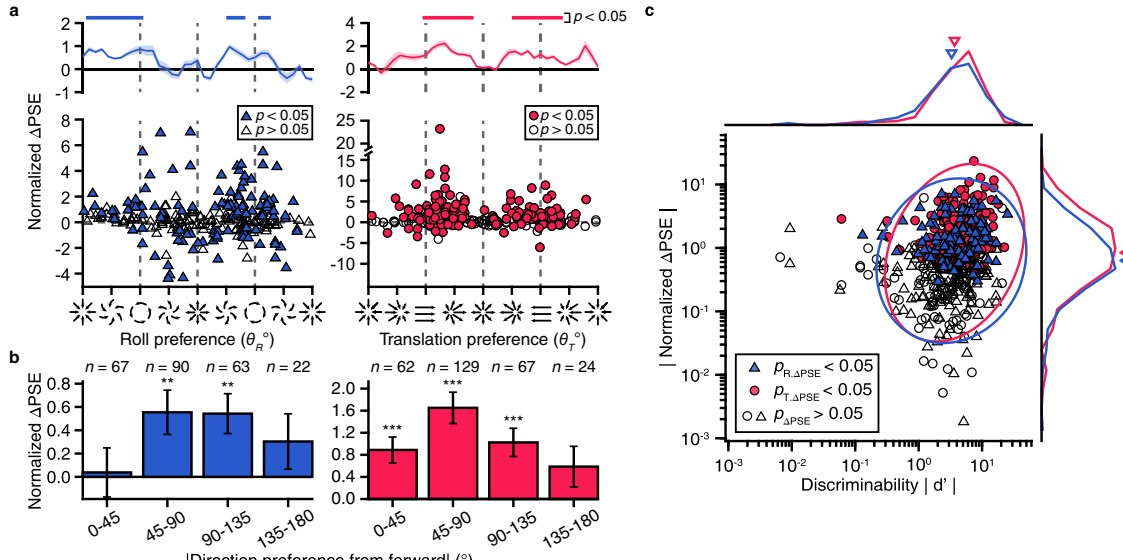

**Fig. 5 | Microstimulation induced PSE shift effect depends on neuronal sensitivity.** Data are similar in fine/coarse version in the two animals and have been pooled together (Supplementary Fig. 5). **a** Relationship between ΔPSE and direction preference. Each point represents data from one stimulated site with filled symbol represents significant ΔPSE ($p < 0.05$, probit regression). Top marginal plots represent ΔPSE as a function of direction preference with a sliding window (step = 10°, bin = 30°). Shaded area, SEM. Superimposed bars indicate significant values different from zero ($p < 0.05$, two-tail $t$-test). **b** Average result in **a** grouped by four relative preferred direction ranges. **$p < 0.01$, ***$p < 0.001$, two-tail $t$-test. In roll task, $p = 0.86$, 4.4e-3, 2.4e-3, 0.22 for four groups, respectively. In translation task, $p = 3.5e-4$, 4.3e-8, 1.6e-4, 0.13 for four groups, respectively. Error bars, SEM. **c** Relationship between ΔPSE and MU discriminability. Colored and open symbols indicate significant and nonsignificant ΔPSE. Data are shown separately for roll (triangle) and translation task (circle). Colored ellipse, 95% confidence ellipses for roll and translation data point, respectively. Top and right panels show marginal distributions of $|d'|$ and $|$ΔPSE$|$ (on log scales). Inverted triangles, geometric means.

In addition to the distance between preferred direction and reference, we also used neural discriminability ($d'$) to assess its impact on microstimulation effect. We ran an analysis of covariance (ANCOVA) to examine dependence of microstimulation effects on $d'$ in both roll and translation tasks (Fig. 5c). Firstly, the analysis revealed a significant slope effect of the regression of $d'$ and induced PSE shift ($p = 4.1e-7$, ANCOVA), illustrating that microstimulation of sites with greater $d'$ tended to produce larger PSE shifts in both tasks. Secondly, the analysis also revealed a significant intercept effect, indicating that after factoring out the dependent variable, the overall PSE shift in the translation task was larger than that in the roll task ($p = 7.0e-4$, ANCOVA).

In sum, microstimulation frequently induced PSE shift in the roll direction discrimination task, particularly for those sites with preferred direction close to CW or CCW. This is because this population of MUs is more sensitive to changes in directions around reference, and thus are potentially more influential to the animals' perceptual judgments.

**Microstimulation biased choice between flow patterns**
As indicated in the above sections, for each trial in the 4-AFC task, animals need to correctly recognize the flow patterns (roll versus translation) in order to discriminate motion directions (CCW versus CW, or rightward versus leftward) by choosing the correct spatial targets (vertical versus horizontal). Recognizing flow patterns is relatively easy when the sensory stimulus is reliable, albeit more difficult for ambiguous stimuli like those close to the reference. Thus, if roll-related signals in MSTd contribute to roll perception, we expect to see microstimulation of a "roll-preferred" site may bias the animals' choice towards the targets for the roll task (vertical meridian) especially on those ambiguous conditions. Analogously, artificially stimulating "translation-preferred" sites may drive the animals to make more choices towards the targets for the translation task (horizontal meridian).

Figure 6a shows this idea with the aid of three toy models. Specifically, Model 1 indicates that microstimulation only biased the monkeys' choice within one of the flow patterns, but does not affect choice in the other. Model 2 indicates that microstimulation biased the monkeys' choice in both flow patterns, with respect to the labeled-line of the stimulated site. Thus, the first two models do not involve interaction of choice between flow patterns. Contrary to model 1 & 2, model 3 predicts that choice is not only biased within one of the flow patterns, but only biased between flow patterns. Figure 6b shows one example of microstimulation experiment consistent with model 2. In particular, microstimulation biased choice not only from CCW to CW in the roll task, but also from leftward to rightward motion in the translation task. Importantly, proportion of choice for each flow pattern was similar between microstimulation and control conditions, indicating no biased choice between flow patterns ($p = 0.52$, two-tail $\chi^2$ test). In contrast, the second example indicates microstimulation result consistent with model 3 (Fig. 6c). In particular, microstimulation biased the animal's choice from rightward to leftward motion in the translation task, and at the same time biased choice from roll to translation task as well ($p = 0.0011$, two-tail $\chi^2$ test). To quantify the effect of biased choice between flow patterns, we calculated a choice-change index (CCI) across ambiguous conditions (see Method). Large CCI values (>0) indicate choice biased towards *translation* task on microstimulation trials compared to the non-stimulated trials, and small values (<0) indicate choice biased towards *roll* task instead. CCI equal to zero indicates unchanged choice frequency between tasks by microstimulation. The first example in Fig. 6b has a CCI value of 0.16, indicating a weak effect of biased choice between flow-patterns. The second example in Fig. 6c however, has a CCI value of 0.74, indicating a strong biased choice effect from roll to translation flow pattern induced by microstimulation.

Next, we examined CCI across population as a function of spiral index that reflected relative strength of tuning preference between roll

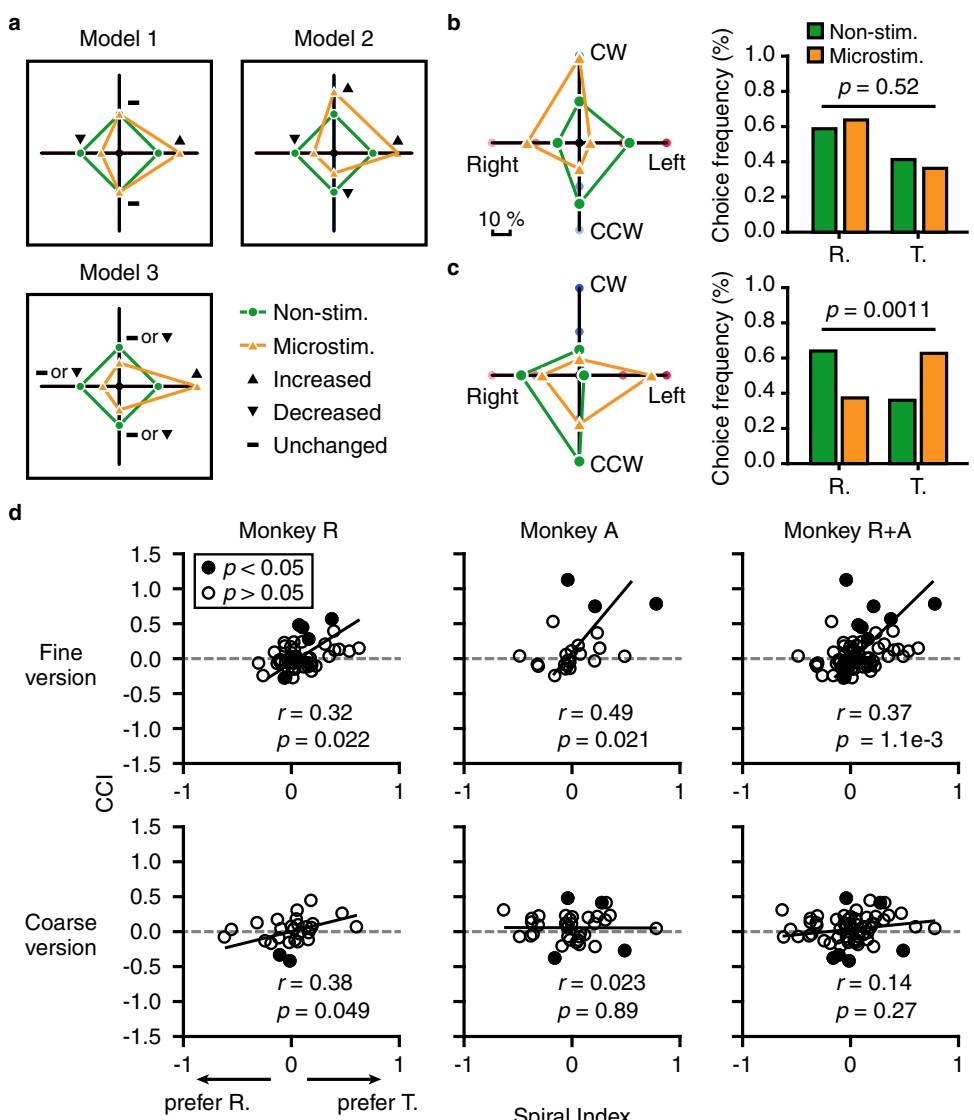

**Fig. 6 | Microstimulation effect on choice between flow patterns. a** Toy models illustrating three possible outcomes: Model 1, choice biased by microstimulation only within one flow pattern; Model 2, choice biased by microstimulation with respect to preferred stimulus within each flow pattern; Model 3, choice biased by microstimulation with respect to preferred stimulus within one flow pattern, and between flow patterns. Orange symbols, choices in microstimulation trial; Green symbols, choices in non-stimulated trials. **b**, **c** Two examples of microstimulation in MSTd. Symbols are same as in **a**. **d** Scatter plot of CCI as a function of spiral index across animals and task paradigms (Spearman rank correlation coefficient). Only CR > 65% for control trials were included in the analysis. Filled circles, significant microstimulation-induced change in choice frequency between roll and translation tasks ($p < 0.05$, two-tail $\chi^2$ test); Black lines, linear regression fit.

and translation flow pattern (Fig. 6d). We found a significant positive correlation between spiral index and CCI for both monkey (monkey R, $p = 0.022$; monkey A, $p = 0.021$, Spearman rank correlation) in the fine version of the task. That is, when a stimulated site more preferred translation motion (spiral index > 0), microstimulation tended to bias inter-flow pattern choice to translation, and vice versa for stimulating sites with the preference for rotation motion (spiral index < 0). Pooling data across the two animals strengthened the significance of this trend ($p = 1.2e-3$, Spearman rank correlation). By comparison, the relationship between spiral index and CCI was weaker in the coarse version of the task, which was reflected by a marginal significance ($p = 0.049$, Spearman rank correlation) in monkey R, and non-significance ($p = 0.89$, Spearman rank correlation) in monkey A. Pooling data across animals failed in increasing significance level. Such difference in the relationship of CCI and spiral index between fine and coarse versions may imply dominance of different models in MSTd. For example, model 3 may largely exist in fine version, whereas model 1 or model 2 may instead dominate in coarse version.

## Microstimulation reduced performance discriminability

In above sections, we have described that artificially activating a population of MSTd neurons by microstimulation induced significant PSE shift in the animals' both roll and translation direction judgment. Microstimulation also biased the animals' choice between flow patterns. These results support that roll signals, as well as translation signals in MSTd are sufficient for the animals' perceptual judgments about the direction of optic flow. In addition to the PSE shift effect, we also noticed a more non-specific deleterious effect in the animals' perceptual discriminability, which may be caused by noise from electrical stimulation[65,79]. The deleterious effect is frequently accompanied with the positive PSE shift effect in previous studies in areas of MT and MST[61,65,73,74,79,80] even when the stimulation current amplitude is as low as 10–20 μA. When the amplitude keeps increasing however, for example, up to 80 μA or higher, the perception-disruption effect becomes more dominant over the specific PSE shift effect[65,79].

We first examined intra-task situations. The example in Fig. 7a showed that microstimulation either decreased CCW versus CW, or

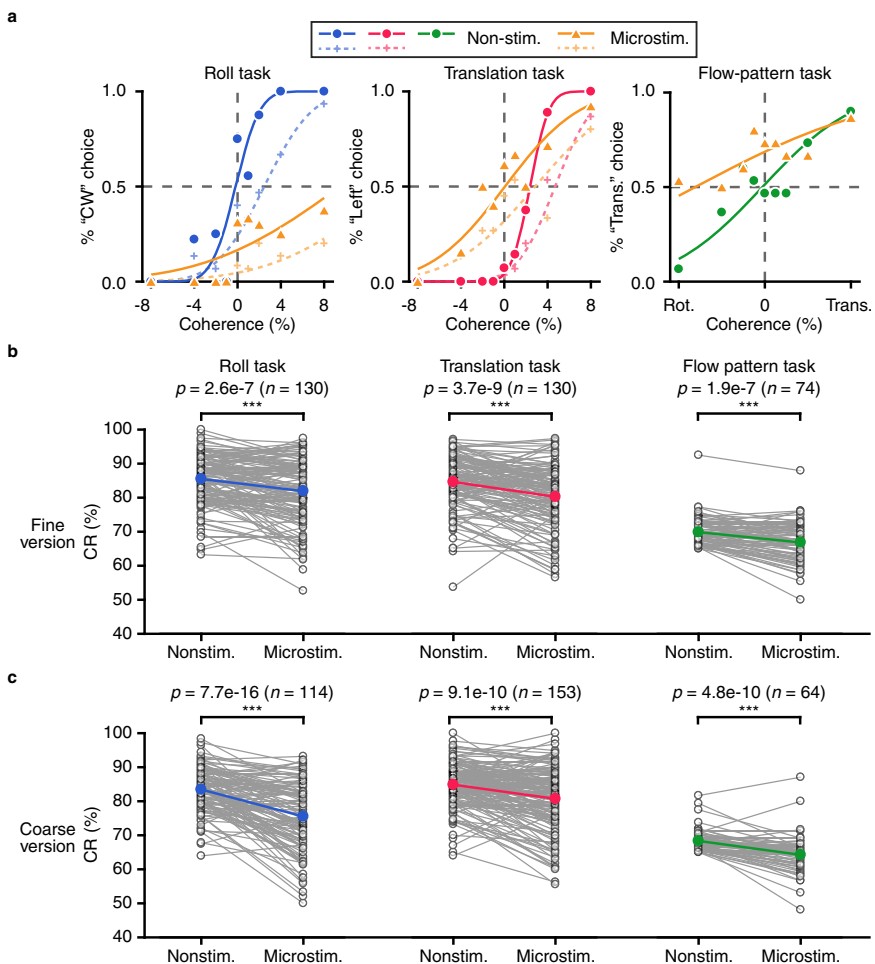

**Fig. 7 | Microstimulation effect on the animals' performance discriminability.**
**a** Example psychometric functions for non-stimulated trials and microstimulated trials. Same format as Fig. 4a. Microstimulation-induced change of CR under fine (**b**) and coarse (**c**) version. Open circles, CR of each individual site with or without stimulated; Colored filled circles, mean CR. Horizontal lines, *p* value for two-tail paired t-test. ***p* < 0.001. Fine-roll, 4.2% in CR reduction (*p* = 2.6e-7, the same hereinafter); fine-translation, 5.2% (*p* = 3.7e-9); fine-flow-pattern, 4.3% (*p* = 1.9e-7); coarse-roll, 9.5% (*p* = 7.7e-16); coarse-translation, 4.9% (*p* = 9.1e-10); coarse-flow-pattern, 6% (*p* = 4.8e-10). For flow-pattern task, only CR > 65% for control trials were included in the analysis.

rightward versus leftward discriminability, as reflected by shallower psychometric functions on microstimulated trials (Fig. 7a, left and middle panels). We then analyzed microstimulation effect on performance discriminability on the same population data that has been analyzed for PSE shift as shown in the previous section (Fig. 4). Across population (Fig. 7b, c), the average correct rate on microstimulated trials was significantly decreased in either roll (fine-roll, *p* = 2.6e-7; coarse-roll, *p* = 7.7e-16; two-tail paired *t*-test), or translation (fine-translation, *p* = 3.7e-9; coarse-translation, *p* = 9.1e-10; two-tail paired *t*-test) task with a modest effective size (4–10%) compared to that on non-stimulated trials.

Microstimulation also deteriorated the animals' inter-flow pattern discriminability. The example in Fig. 7a (right panel) showed that on microstimulated trials, the animal made more incorrect choice with respect to the flow pattern. Across population, microstimulation decreased the inter-flow pattern correct rate with a modest albeit significant change of 4–6% (fine: *p* = 1.9e-7, coarse: *p* = 4.8e-10, two-tail paired *t*-test; Fig. 7b, c, right panels). This effective size was similar to that for intra-flow pattern discrimination tasks, and was consistent across animals (Supplementary Fig. 6a).

The non-specific deleterious effect in the animals' perceptual tasks raises a potential concern that whether it may interfere with the more specific PSE shift effect, particularly for those "unexpected" cases when PSE shift was in the anti-preferred direction of the stimulated

site. We think this is unlikely due to a number of facts. First, the deleterious effect was robust, yet small (4–10% in CR reduction), suggesting that its impact on perception is limited in our study using current amplitude of 20–40 μA. Second, by comparing results between different current amplitude (20 μA vs. 40 μA, Supplementary Fig 4), we found that in general the larger current produced slightly stronger effect in both of the deleterious and PSE shift effects compared to the smaller current. Importantly, the larger PSE shift was still in the expected direction with respect to the labeled-line of the stimulated site, indicating the non-specific effect cannot account for the negative PSE shift cases in our data, i.e., roll and translation direction discrimination tasks. Third, plotting microstimulation-induced PSE shift against CR change on a site-by-site basis (Supplementary Fig. 6b), we found that the negative PSE shift cases did not necessarily happen in those sessions with poorer behavioral performance.

Hence, based on our analyses, we infer that the non-specific deleterious effect does not interfere with the specific PSE shift effect, but rather, it may imply noise introduced due to currents being spread into neighboring clusters with inconsistent neuronal features. Such an effect may be used to imply a necessity causal role of the manipulated areas. However, other possibilities including distraction of attention due to direct detection of electrical currents, or activation of passing fibers, should be further tested to explore the exact sources of such effects.

## Comparison of microstimulation effect in 2-AFC task

In the 4-AFC task, the two sources of choice errors (i.e., intra- and inter-flow pattern) coexisted and may interfere with each other. In our study, one animal (monkey R) completed a simpler 2-AFC paradigm for roll task and translation task when the two flow patterns were run in separate blocks, providing a chance to test the intra-flow pattern discriminability without any inter-task interference. In particular, the overall experimental paradigm was similar to the 4-AFC task, except that only two vertically arranged choice targets (for 2-AFC roll task) or two horizontally arranged choice targets (for 2-AFC translation task) were presented on the display in an experimental block. After training, monkey R achieved stable performance in both fine (Supplementary Fig. 7a, left 3 panels) and coarse versions (Supplementary Fig. 7a, right 3 panels) of 2-AFC task. Overall performance in the translation task was better than that in the roll task because this animal had repeatedly performed the translation task in previous studies.

We found microstimulation frequently induced significant PSE shift in 2-AFC tasks (Supplementary Fig. 7b). Importantly, the magnitude of shift was indistinguishable from that in the 4-AFC task ($p > 0.05$, two-tail non-paired $t$-test, Supplementary Fig. 7c). Similarly, microstimulation also significantly disrupted the animal's perception in 2-AFC task with an effective size comparable to that in the 4-AFC task (Supplementary Fig. 7d, e). Thus, microstimulation effect on direction discriminability for each flow pattern, is similar between the two types of tasks, indicating that PSE shift in 4-AFC task in each flow pattern task is not affected by inter-flow pattern discrimination error. This is consistent with the result when comparing PSE shift based on trials with only correct flow pattern choice, with that based on trials including choice towards the wrong flow pattern (Supplementary Fig. 2b).

## Independence of roll perception on translation component

Roll and translation signals are two elementary first-order flow components that are supposed to be orthogonal to each other[81]. However, due to aperture problem, it is likely that neural activity observed from one type of stimuli is actually evoked by the other within a restricted visible field. MT neurons, for example, suffer from aperture problem due to their much smaller receptive fields compared to that in MSTd, and thus have been proved not to encode complex optic flow per se[53]. In our current study, although we have recorded MUs from MSTd, and many of them contain classic, large receptive field that invade into the ipsilateral field, there are still cases when receptive fields are relatively restricted, and hard to be distinguished from MT due to anatomical continuum in brain. In addition, we did not perform position-invariant measurements as in previous studies[33,53,82]. Thus, we performed an analysis by examining similarity of preferred flow pattern within the receptive field of each site being activated by microstimulation (Fig. 8). Figure 8a shows the same example site as in Fig. 2a, with quite different preferred motion vectors within its receptive field. To quantify this, we calculated a similarity index which was defined as the proportion of vectors with angle <45° between roll and translation within the receptive field. The similarity index ranges between 0 and 1, indicating different (perpendicular to each other or even in opposite directions) and similar preferred motion vectors, respectively (see more examples in Fig. 8b). We found that across population, both similar and dissimilar cases co-existed in MSTd, generating a continuous distribution (Fig. 8c). Furthermore, we found that consistent with intuition, sites with similar preferred motion vectors tended to carry smaller and more peripheral receptive fields (Fig. 8d), raising the concern that roll and translation modulations may be correlated for those sites vulnerable to aperture issue. However, microstimulation on sites with low similarity index tended to shift PSE in roll task even more than that on sites with high similarity index (Fig. 8e), contrary to predictions from the translation-origin hypothesis. In addition, sites with RFs containing fovea also tended to show significant PSE shift (Fig. 8f). These results indicate that microstimulation effects in roll tasks cannot largely be explained by the translation component.

## Discussion

In the current study, we found MSTd MUs are predominantly "spiral" by containing both roll and linear translation components. Through microstimulation techniques applied in a newly designed 4-AFC roll and translation direction discrimination task, we provide direct evidence, for the first time to our knowledge, showing that the roll-related signals causally contribute to rotation perception about the line-of-sight axis. This effect is analogous in the fine and coarse version of the tasks, indicating that roll-related signals could either directly contribute to high-order stimuli (spiral) discrimination, or could be extracted for low-order stimuli (pure roll) discrimination. Our study thus fills in an omission by providing a more complete picture about functional implications of complex flow patterns in MSTd to direct perception.

Although roll and translation are two elementary flow components, the combination of them, instead of each of them, is more frequently present in MSTd as found in our current study based on MU (Fig. 3), as well as based on single-unit (SU) data in previous study[33]. Some researchers have used this fact to argue that MSTd does not use decomposed elementary flow elements such as translation for navigation-related perception because otherwise there would be more translation-only or rotation-only neurons[33]. In another case, Orban found weaker modulations when translation stimulus was superimposed with rotation stimuli, suggesting that low-order, elementary flow components are unlikely to be extracted from higher-order stimuli[53]. However, our experiments clearly demonstrate that spiral MUs do contribute to roll and translation direction judgments. In particular, we show that in the fine experimental version, microstimulation of spiral MUs significantly affects the first-order translation, as well as the higher-order spiral direction judgments. In the coarse version, microstimulation also significantly influence perceptual judgments in both first-order translation and pure roll tasks. Our results are consistent with a recent study showing that activities from spiral MUs in MSTd are covaried with the monkey's perceptual judgment about translation direction on a trial-by-trial basis[57]. However, our results are inconsistent with another study which fails to find any significant choice correlations in a coarse flow-pattern discrimination task[72]. We argue that functional implications of choice correlation are under debate because this measurement is easily confounded by signals other than sensory readout[59], such as top-down feedback[83] or correlated variability among single neurons[84]. Thus our current study by using microstimulation, provides a more direct insight into whether and how MSTd activities are deployed for perceptual decision making tasks.

Although we reported here that microstimulation significantly influenced the animals' roll perceptual judgment, we noted that cases of PSE shift in the "unexpected" direction were relatively more frequently present (30–35%) in roll tasks compared to that in the translation tasks (15–20%, Fig. 4b, Supplementary Fig. 3c). Here we discussed a few possibilities that may mediate this heterogeneity in microstimulation effects between the two tasks. Previous studies have indicated that under conditions when neurons are causally involved in perceptual judgments, the actual microstimulation effects are limited by a number of factors including electrical-stimulation parameters[79], clustering, and neuronal sensitivity relative to the task context[61,73,74]. In our current study, we have used typical electrical-stimulation parameters that allows to best see microstimulation effect. We also show that roll signals in MSTd are clustered as well as translation signals. As we have purposely chosen the sites with high clustered structure to apply microstimulation, it is unlikely that it can explain difference in microstimulation effects across different experimental conditions. Note though, the current clustering index was based on measurement

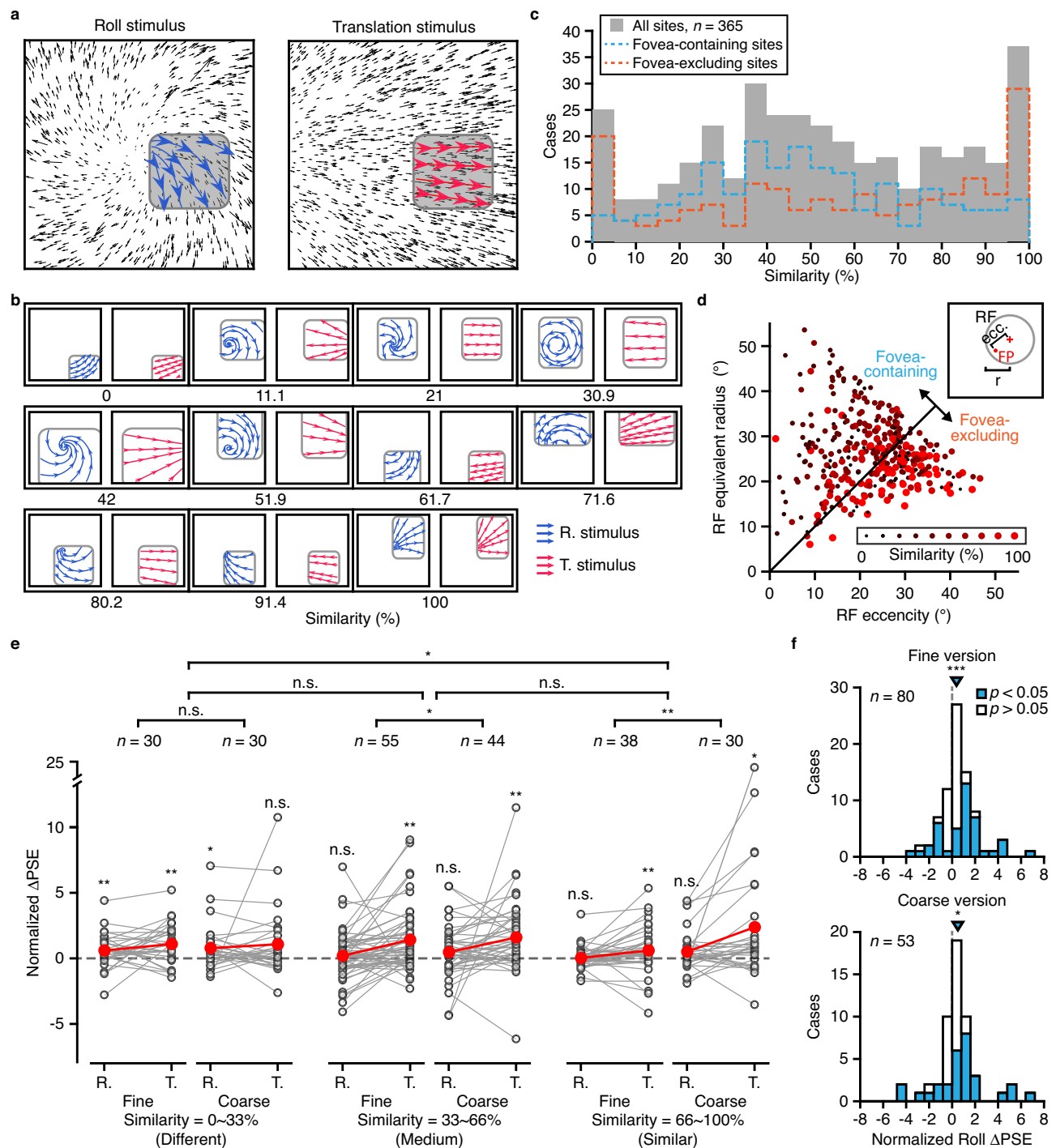

**Fig. 8 | Similarity of preferred optic flow pattern within the receptive field.**
**a** The preferred optic flow pattern in roll-rotation plane and translation plane for the same example site as in Fig. 2a. Gray square, the receptive field of the MU. Blue and red quivers, the vector flows in RF for roll-rotation stimuli and translation stimuli, respectively. Similarity of preferred optic flow pattern for this MU is 37.0%. **b** More examples of preferred optic flow pattern in RF for different similarity index. **c** Continuous distribution of similarity ranging from 0% (different) to 100% (similar). Gray bar, all data; Blue line, sites with RF containing the fovea; Orange line, sites with RF without covering fovea. **d** Relationship between similarity and RF properties. Similarity (color and size of dot) is plotted as a function of RF equivalent radius and eccentricity. **e** Normalized microstimulation-induced PSE shift under roll and translation task across three groups of similarity. Open dots, ΔPSE for individual site; Red dots, median of ΔPSE. Asterisk above dots, median ΔPSE significantly different from zero (n.s. not significant; *$p < 0.05$; **$p < 0.01$, two-tail sign test). Different group: $p_{\text{fine-roll}} = 5.2\text{e}{-}3$, $p_{\text{fine-translation}} = 5.2\text{e}{-}3$, $p_{\text{coarse-roll}} = 0.043$,

$p_{\text{coarse-translation}} = 0.099$; Medium group: $p_{\text{fine-roll}} = 0.28$, $p_{\text{fine-translation}} = 2.7\text{e}{-}3$, $p_{\text{coarse-roll}} = 0.096$, $p_{\text{coarse-translation}} = 1.3\text{e}{-}3$; Similar group: $p_{\text{fine-roll}} = 0.87$, $p_{\text{fine-translation}} = 5.1\text{e}{-}3$, $p_{\text{coarse-roll}} = 0.099$, $p_{\text{coarse-translation}} = 0.016$. Horizontal lines, $p$ value for 2-dimensional Kolmogorov–Smirnov test between fine and coarse versions or different groups of similarity. n.s. not significant; *$p < 0.05$; **$p < 0.01$. Fine version vs. coarse version: different group, $p = 0.084$; medium group, $p = 0.025$; similar group, $p = 1.1\text{e}{-}3$. Different group vs. Medium group, $p = 0.17$; Medium group vs. Similar group, $p = 0.088$; Different group vs. Similar group, $p = 0.020$. **f** Distribution of ΔPSE under roll task for sites with RF containing fovea in the fine (top) and coarse (bottom) versions. Filled and open bars represented significant ($p < 0.05$, probit regression) and nonsignificant ($p > 0.05$, probit regression) ΔPSE, respectively. Inverted triangles, median ΔPSE; Asterisks, median ΔPSE significantly different from zero; Fine version, $p = 4.5\text{e}{-}4$; Coarse version, $p = 0.013$. Dashed vertical lines: zero ΔPSE.

along each electrode penetration (i.e., one-dimension), thus we do not know how neuronal signals are actually clustered around the stimulation site in a 3-dimensional volume. It is likely that more complex optic flow patterns are less clustered in 3D compared to the translation signals in MSTd. Future studies by using electrode array may provide better estimate of clustering structure in the sensory cortices.

Other than clustering, factor of neuronal sensitivity has also been shown to be tightly related to microstimulation effect. For example, Britten and van Wezel found that cases with PSE shift in unexpected direction tend to happen when stimulation sites exhibit weaker tuning[74,85]. By applying analysis of covariance (ANCOVA), we found two facts. First, microstimulation effects are significantly dependent on neuronal sensitivity. That is, more sensitive neurons (MUs), the higher chance to see large and significant microstimulation induced PSE shift in the animals' psychometric functions. This may be one reason why overall microstimulation effects appear to be a bit weaker in the roll task compared to that in the translation task (Fig. 4), because overall roll tuning is slightly weaker than translation. Second, ANCOVA also revealed a significant intercept effect, indicating that even after factoring out the regression issue, the average microstimulation effect for roll task was still weaker compared to translation. This result may suggest that decoding information along the rotation flow dimension is more complicated and heterogeneous than along the translation dimension. Thus, if rotation signals are employed by the brain for multiple functions, it is likely that cases with PSE shift in the opposite direction would be more frequently encountered when restricting the animals to perform direction discrimination within the roll context.

In addition to comparison of microstimulation effect between roll and translation tasks, we also compare two versions of task: fine and coarse that differ dramatically in the way of controlling task difficulty. Specifically, in coarse task subjects need to distinguish two largely deviated motion directions (e.g., left vs. right) with the difficulty controlled by visual coherence varied from high to as low as 0%[86]. By contrast in the fine task, visual coherence is fixed at a certain level, whereas task difficulty is controlled by fine steps of motion directions deviated away from the reference[56,78]. It is unclear whether the two tasks involve same or different sensory readout mechanisms because the two tasks have rarely been applied on the same neuron, same animal under similar experimental conditions. A previous human psychophysical study reports heterogeneous perceptual bias effect between the two tasks, suggesting different sensory readout mechanisms[87]. In our current study, we run both fine and coarse tasks on many of the same sites, allowing to directly compare decoding effects as assessed by microstimulation-induced PSE shift between the two tasks on a site-by-site basis. We show they are comparable without statistical-significant difference for both roll and translation direction discrimination (Fig. 4c, Supplementary Fig. 5), indicating that the two task contexts may share the same sensory readout mechanisms. A key factor leading to difference between our results and that from previous psychophysical studies may lie in the assumption about which population of neurons contribute most to the fine context[87,88]. In particular, researchers have suggested that neurons with preferred direction slightly away from the reference (e.g., 40–60°) contribute most to fine discrimination, which differs from those neurons with nearly opposite direction preference (e.g., ±90°) that matter most in coarse discrimination. However in many areas, neurons can have fairly broad tuning width[52,89,90], so that neurons with preferred directions quite far away from the reference could contribute most for both fine and coarse contexts at the same time. For example, previous studies demonstrate that MSTd is dominant by neurons with leftward and rightward preference (i.e., -90° away from straight forward), and these neurons are highly sensitive to heading directions varied in fine steps around straight ahead[55]. Thus both fine and coarse task may employ the same population of neurons with similar weight for perceptual decision making, as seen in our current data (Supplementary Fig. 5).

However, we do see that inter-task (roll versus translation) microstimulation effects are different between the fine and coarse versions. Specifically, in the fine version, CCI induced by microstimulation is more dependent on the relative preference of roll and translation stimuli of the stimulated site, whereas this dependence is less clear and consistent across animals in the coarse version. Thus it is likely that decoding mechanism for different flow patterns is not same under the fine and coarse contexts.

MSTd has largely been hypothesized to be involved in self-motion perception for a number of reasons. First, its neurons contain much larger visual RFs compared to those in upstream areas such as MT[35,46,91]. The large RFs are suitable for integration of global optic flow information which is highly relevant to self-motion[1-6]. Second, MSTd neurons are modulated by focus of expansion of optic flow that is varied away from the center of the visual display, simulating different "heading" directions during spatial navigation[54]. Third, in addition to visual signals, MSTd contains a large population of neurons carrying robust vestibular signals with aligned preferred directions with those simulated from optic flow, a phenomenon of which is prevalent across a number of cortical regions such as VIP, VPS, FEFsem (see review by Gu, 2018[92]). These neurons typically show enhanced activity during congruent bimodal stimulation, providing an ideal neural correlate for self-motion perception during natural locomotion or navigation. Note though, that MSTd also contains another large population of incongruent neurons with nearly opposite heading preference based on vestibular and that simulated from optic flow. It has been hypothesized that different populations of neurons in MSTd with either visuo-vestibular congruent or conflict relations could be used to segregate independent object-motion from self-motion[93-97].

However, alternative hypothesis also exists. That is, rotation and translation signals in MSTd, and as in many other visual regions, may be responsible for 3D visual perception. For example, although RFs of MSTd neurons are usually large, they seldom cover the whole visual field that is typically accessible from the two eyes. In addition, tuning functions of many units show spatial invariance for complex optic flow pattern such as roll, suggesting that at least on single neuron level, it is unlikely to extract reliable focus of expansion by MSTd itself[33]. In this case, it is likely that vestibular signals may be used for updating visual stability during grabbing or smooth eye pursuit[98] when self-motion happens at the same time.

In sum, to our knowledge there is so far no direct evidence which is able to pin down functions of MSTd in self-motion and object-motion perception. Future studies are required to address this issue to fully understand the role of MSTd, as well as other regions with similar properties in brain. In any case, using microstimulation techniques, our study provides direct evidence supporting that roll signals, as well as translation signals in MSTd, are indeed decoded by the brain, in a way accordant to their coded labeled-lines, for perception of rotation with respect to gravity and linear translation motion, respectively. This motion could arise either from external objects, or from ego-motion in the environment.

## Methods
### Subjects
A total of two healthy male rhesus monkeys (*Macaca mulatta*, monkey R and monkey A, weighing 9-10 kg) were included in this study. Before the study, the animals were chronically implanted using stereotaxic apparatus with a circular molded, lightweight plastic ring (Monkey R: chamber center anterior-posterior, AP = + 0.4 cm, medial-lateral, ML = 0 cm, inner diameter = 5.5 cm; Monkey A: chamber center AP = + 0.6 cm, ML = 0 cm, inner diameter = 6 cm) through the titanium inverted T-bolts and dental acrylic anchored to the skull for head restraint. The rings served for both head-fixed post and neural recording chamber. After recovery, the animals were trained to seat in a custom-built primate chair with their heads restrained and perform

the behavioral tasks in the visual virtual reality system[59]. All animal procedures were approved by the Animal Care Committee of Shanghai Institutes for Biological Science, Chinese Academy of Sciences.

## Apparatus and visual stimuli

The visual virtual reality system consists of an LCD monitor (HP LD4710, 60 Hz, 1280 × 1024) that is ~32 cm of viewing distance in front of the animals, covering ~90° × 90° of visual angle. To simulate the visual optic flow of real motion, visual stimuli are depicted as the projected image of a "camera" moving though a 3D star cloud. The star cloud occupies a virtual space for $100 × 100 × 40$ cm that is uniformly filled with 4 pixel yellow dots with a density of 0.01 dots cm$^{-3}$ in dark background generated by OpenGL accelerator board (NVidia Quadro 2000). The display contained a variety of depth cues, including motion parallax and size information but without binocular disparity. The animals viewed visual stimuli binocularly, yet no horizontal disparity information per se was provided in the visual stimuli to indicate motion in depth. Thus, there was a cue conflict between binocular information of the visual display and the motion information. Motion coherence was manipulated by keeping a certain percentage of dots moving coherently while the remaining dots randomizing the 3D location on each frame[59]. The visual stimuli were controlled by customized C + + software and synchronized with the electrophysiological recording system by TEMPO (Reflective Computing, U.S.A).

## Experimental paradigm

**Fixation and tuning measurement**. The monkeys were required to simply maintain fixation on a red point (0.5° × 0.5°) at the center of the screen during the stimulus duration until the fixation point was extinguished at the end of trial indicating reward of water or juice. The monkeys' gaze direction was typically required to be restricted within an invisible electronic window (2° × 2°) around the fixation point.

We used complex optic flow patterns to characterize MSTd cells' responses in a 3D spiral space[33,57]. Specifically, radial motions (forward and backward), were combined with rotational motions around line-of-sight (CW or CCW), which defined a roll plane. The radial motions were also combined with lateral translation motions (leftward or rightward), leading to varied focus of expansion that defined a translation plane. The roll and translation planes were perpendicular to each other that shared the same axis of forward and backward radial motion. The roll and translation stimuli were always respectively presented in different trials without being combined together on a single trial. All visual stimuli in the tuning measurement experiment contained 100% coherence. Finally, a null condition without any optic flow stimuli was served as a condition to measure the spontaneous activity. All stimulus conditions were presented in a pseudorandom order with each trial lasting 1000 ms. Each stimulus conditions were repeated at least three times in each experimental session.

**Four-alternative forced choice (4-AFC) combined roll-translation discrimination task**. In the main experiments, the animals were trained to perform a 4-AFC optic flow discrimination task. In each trial, the animals initiated a trial by fixating on a fixation point appeared at the center of the visual display. After fixation, four choice targets (0.5° × 0.5°) appeared symmetrically around the fixation point by 5 degrees (top, down, left and right). The two vertically aligned targets represented choice for roll-rotation discrimination task (CCW and CW, respectively), and the two horizontally aligned targets represented choice for the translation direction discrimination task (rightward and leftward, respectively). Optic flow stimuli appeared at the same time, lasting 1000 ms before the fixation point disappeared indicating a go-signal to make saccadic choice. All stimulus conditions were randomly interleaved on a trial-by-trial basis in each experimental session. Thus in this task, the animals were required to make correct judgments both about flow pattern and motion direction with a guess rate of 25%.

Reward was delivered at the end of trial if the animals' choice matched the visual stimuli.

**Two-alternative forced choice (2-AFC) separated roll or translation task**. In addition, one of the animals (monkey R) completed a simpler 2-AFC task. The overall experimental paradigm was similar to the 4-AFC task. However, two separate sessions were conducted with one block of roll-rotation task and one block of translation task. In each block, only two choice targets were presented to indicate motion directions in each task with a guess rate of 50%. The two blocks were presented in different order across days.

**Fine task version**. Both 4- and 2-AFC tasks were performed under a fine block and a coarse block. In the fine version, task difficulty was manipulated through a rotary or translational angle deviated away from the reference of straight forward motion ($\theta_R = 0°, \theta_T = 0°$). In particular, a small CCW ($-\theta_R$) or CW ($+\theta_R$), or a small rightward ($-\theta_T$) or leftward ($+\theta_T$) motion component was introduced with respect to the reference in the roll and translation discrimination task, respectively. Thus across trials, the rotational component ($\theta_R$) or translation component ($\theta_T$) were varied in fine steps around the reference of straight forward. In roll task, eight $\theta_R$s plus a reference were included in monkey R (±11.54°, ±5.74°, ±2.87°, ±1.15°, 0°), and in monkey A (±20°, ±10°, ±5°, ±2.5°, 0°). In translation task, eight $\theta_T$s plus the reference were included in monkey R (±5°, ±2.4°, ±1.15°, ±0.55°, 0°) and monkey A (±7°, ±3.5°, ±1.75°, ±0.88°, 0°). In the 4-AFC task, the reference was shared by both the roll and translation task, so the number of total stimulus condition was 17. For the 2-AFC task, there were 9 stimulus conditions for either the roll or translation task. In all cases, visual coherence of the optic flow was fixed at ~13% for both monkeys. Each stimulus conditions were presented with 10-15 repetitions to build up psychometric functions.

**Coarse task version**. Unlike the fine version, task difficulty was instead controlled through visual coherence. The reference was a 0% coherent flow field that contained no any clear information. In roll, visual motions only contained rotational component ($\theta_R = \pm 90°$), with coherence of the flow was varied between [±8%, ±4%, ±2%, ±1%, 0%] for both animals. In translation task, visual motion only contained laminar motion ($\theta_T = \pm 90°$), with coherence also varied within [±8%, ±4%, ±2%, ±1%, 0%]. Fine and coarse versions were run in different blocks in different order across days.

## Electrophysiology

**Recordings**. We recorded extracellular MU activities by single tungsten electrodes (FHC, impedance ~500 kΩ) or platinum iridium electrodes (FHC, impedance ~ 500 kΩ) through a tightly settled, horizontally placed recording grid inside the plastic ring. Each penetration was advanced into the cortex through a trans-dural stainless steel guide tube using a micromanipulator (FHC). Electrophysiological data were collected from two hemispheres in two monkeys (both left hemisphere for monkey R and monkey A).

The dorsal part of medial superior temporal area (MSTd) was targeted with aid from structural MRI data collected on the Brain Function Imaging Platform of Center for Excellence in Brain Science and Intelligence Technology (3 T MRI scanner, Siemens). During scan, a few tungsten electrodes were penetrated to the region of interest according to the stereotaxic coordinates. MSTd area could be identified based on the relative location of the reference electrode artifacts in the brain (Supplementary Fig. 1a). In subsequent experiment however, MSTd area was also cross-validated by physiological properties, including gray/white matter pattern, size of receptive field (RF), responses to visual stimuli and direction selectivity for visual motion[46,68,99–101]. RFs of the isolated MUs were hand-mapped with a bar stimulus of flashing and drifting random dots moving in preferred

directions around the visual field. Based on the mapping results, area MSTd was distinguished by its spatial location with respect to the adjacent middle temporal area (MT) and the lateral part of MST (MSTl). In particular, MSTd typically locates on top of MT when penetrating electrode perpendicular to the earth-horizontal plane, whereas MSTl typically locates more anterior, lateral and ventral, without MT underneath. In addition, MSTd neurons typically contained larger RFs that occupied a quadrant to a full hemifield, and frequently covered the fovea and some part of the ipsilateral visual field. In contrast, MT and MSTl neurons typically contained more restricted RFs that rarely went beyond the fovea and the ipsilateral field. Plus, MT and MSTl neurons are very sensitive to very small stimulus patch (~1° × 1°), and usually contained strong inhibitory surround that will be suppressed for large visual stimuli.

**Microstimulation procedures.** Before microstimulation, each site of MU activity modulated by optic flow defined in the spiral space was first measured. Tuning curves were measured online for at least three consecutive recording sites apart by ~100 μm. Two conditions were required: (1) modulations to optic flow was significant ($p < 0.05$, one-tail Wilcoxon rank-sum test with spontaneous activity), and (2) tuning curves were similar for at least two consecutive sites in either the roll-rotation plane or translation plane. Across our recordings, most of the time, the three consecutive sites exhibited significant and similar tuning, yet there were some cases when only two consecutive sites exhibited significant and similar tuning whereas the third site lacked clear tuning. In this case, the electrode was retracted back to the middle site of the three consecutive sites for applying microstimulation with a weak current (20 or 40 μA, 200 Hz, biphasic, cathodal-anodal square wave, pulse width = 200 μs, interpulse delay = 100 μs, Alpha Omega SnR, Israel). The stimulation lasted 1000 ms as the visual stimulus duration within a trial. Microstimulation was applied randomly in one half of the trials in each block of the behavioral task. Note that the reward was always delivered based on the animals' correct response for visual stimulus alone, without consideration of whether and how microstimulation was applied. In some cases, neuronal tuning curves were re-measured after microstimulation to confirm the stability of visual modulation at the stimulated sites. Electrical current with 20 μA was initially used for a few sessions, and then a larger current of 40 μA was used throughout the rest sessions, the latter of which occupied majority of the cases. The larger current produced slightly larger effect on the animal's behavioral performance compared to the weaker one, yet directions of the effect were consistent between the two conditions. Data across the two current amplitudes were thus pooled to increase statistical power.

### Data analysis

**Cell type classification.** We divided all the recorded MUs into four groups according to one-way ANOVA test of tuning curves in the roll and translation plane. MUs that are modulated only in the roll plane but not translation plane ($p_{Roll} < 0.001$, $p_{Translation} > 0.001$, one-way ANOVA) was defined as roll-only group. MUs that are modulated only in the translation plane but not roll plane ($p_{Translation} < 0.001$, $p_{Roll} > 0.001$, one-way ANOVA) were classified as translation-only group. MUs with significant modulations in both planes are defined as spiral MUs ($p_{Roll} < 0.001$, $p_{Translation} < 0.001$, one-way ANOVA). Note that when tested with ANOVA, the two radial directions of pure expansion and contraction were dropped out. A final cell type, untuned MUs, consisted of MUs without any significant modulations ($p_{Roll} > 0.001$, $p_{Translation} > 0.001$, one-way ANOVA).

**Spiral index.** Many MUs are modulated by both roll and translation stimuli, yet the relative preference for each type of stimuli is varied among sites. To assess this, we first computed a response vector defined by both rotation and translation stimuli in the 3D spiral space.

Based on this, we then computed a spiral index defined as

$$\text{Spiral Index} = \frac{\cos(\alpha) - \cos(\beta)}{\cos(\alpha) + \cos(\beta)} \quad (1)$$

Where $\alpha$ represents the angle between the response vector in 3D spiral space and the translation plane, whereas $\beta$ represents the angle from the 3D response vector to the roll plane. The spiral index ranges between $[-1 +1]$, with $-1$ indicating strong roll preference and $+1$ indicating strong translation preference.

**Clustering.** Clustering was described as the similarity between tuning curves recorded at adjacent locations along a penetration apart from 100 μm, 200 μm and 300 μm. Strength of clustering was assessed quantitatively by Pearson's correlation coefficients of tuning curves measured at each pair of these sites. Coefficients of $-1$ indicated opposite tuning similarity and $+1$ indicated same tuning similarity.

**Tuning strength.** The strength of roll selectivity or translation selectivity of MU activity was quantified using $d'$, computed as[62]

$$d'_R = \frac{(R_{CCW} - R_{CW})}{\sqrt{\frac{\sigma^2_{CCW} + \sigma^2_{CW}}{2}}} \quad (2)$$

$$d'_T = \frac{(R_{right} - R_{left})}{\sqrt{\frac{\sigma^2_{right} + \sigma^2_{left}}{2}}} \quad (3)$$

where $R_{CW}$, $R_{CCW}$, $R_{left}$, $R_{right}$, $\sigma_{CW}$, $\sigma_{CCW}$, $\sigma_{left}$, and $\sigma_{right}$ represent the mean responses and standard deviations (SD) to stimuli at $\theta_R = \pm 90°$ (CW and CCW) or $\theta_T = \pm 90°$ (leftward and rightward), respectively. The $d'$ shows how well neural activity discriminates between CW and CCW roll or leftward and rightward translation. An absolute value of $d'$ close to 0 indicates that neural activity is roughly equal for $\pm 90°$, thus indicating weak tuning and discriminability between the two stimuli. The larger value, either positive or negative, indicates strong tuning and discriminability between the two stimuli.

**Psychophysics.** To quantify the behavioral performance in each discrimination task, we constructed psychometric functions. In particular, roll psychometric functions were plotted from the proportion of "CW" choices as a function of the "rotation axis", which was defined as the rotational component $\theta_R$ in the fine version, or coherence in the coarse version. In the 4-AFC task, the animals might saccade to the two horizontal targets that correspond to translation task, thus these trials were excluded when building up the roll psychometric functions. Analogously, translation psychometric functions were plotted from the proportion of "Left" choices as a function of the "translation axis", which was defined as the translation component $\theta_T$ in the fine version or coherence in the coarse version. Similarly, trials with choice to the vertical targets corresponding to roll task were exclude in this case. To give a general error level for both roll task and translation task in the 4-AFC paradigm, we also constructed additional psychometric functions based on corresponding choices among four alternative targets. A flow-pattern psychometric function could also be constructed from the 4-AFC data. Specifically, proportion of "translation" choices (the two horizontal targets) was plotted as a function of a flow-pattern axis. The left side of the flow-pattern axis represents roll, with all the absolute values of $\theta_R$, or coherence being normalized to $-1$. The right side of the flow-pattern axis represents translation, with all the absolute values of $\theta_T$ or coherence normalized to $+1$. The middle point of the axis is 0, indicating $\theta_R = 0°$ and $\theta_T = 0°$ in the fine version or 0% coherence in the coarse version, which is the ambiguous condition for inter-task discrimination.

The normal psychometric functions:

$$\text{proportion of "CW"} = \frac{CW}{CW + CCW} \quad (4)$$

$$\text{proportion of "Left"} = \frac{Left}{Left + Right} \quad (5)$$

$$\text{proportion of "Translation"} = \frac{Left + Right}{Left + Right + CW + CCW} \quad (6)$$

The general error psychometric functions:

$$\text{proportion of "CW"} = \frac{CW}{Left + Right + CW + CCW} \quad (7)$$

$$\text{proportion of "Left"} = \frac{Left}{Left + Right + CW + CCW} \quad (8)$$

where $CW$, $CCW$, $Left$, and $Right$ represent the number of the corresponding choices. Psychometric functions were fitted with cumulative Gaussian functions[59] that included two free parameters: (1) point of subjective equality (PSE) corresponding to the value of motion axis that produced equal probability of two choices for the normal psychometric functions or equal probability of four choices for the general error psychometric functions, and (2) threshold taken as one standard deviation of the fitted Gaussian function which corresponded to 84% correct rate. In microstimulation experiments, the significance of changes in either parameter were assessed by a probit regression[61,74].

**Normalizing microstimulation effects across tasks.** To compare microstimulation effects between tasks with different quantity (e.g., rotary angle vs. varied focus of expansion along horizontal meridian, fine vs. coarse), the induced PSE shift was divided by the psychophysical threshold of the non-stimulated psychometric function[59,77] as

$$\text{Normalized PSE shift} = \frac{Induced\ PSE\ shift}{Threshold_{non\text{-}stim.}} \quad (9)$$

As a result, the "Normalized PSE shift" is unitless and can be compared directly across flow patterns (roll vs. translation), and fine vs. coarse version.

**Choice change index (CCI).** In order to quantify the microstimulation effects on monkeys' choice frequency for different flow patterns, we calculated the choice change index as:

$$\text{CCI} = \frac{Choice_{stim.} - Choice_{non\text{-}stim.}}{Choice_{non\text{-}stim.}} \quad (10)$$

where $Choice_{stim.}$ and $Choice_{non\text{-}stim.}$ represents the frequency of monkeys' choices for translation task (targets on horizontal meridian) during the ambiguous conditions in simulated trials and non-stimulated trials, respectively. Here, the ambiguous conditions included the reference of straight forward motion in the fine version, or 0% coherence in the coarse version. In addition, adjacent stimulus conditions in which animals also frequently made wrong choice for flow patterns were also included to increase statistical power (Monkey R: ±1.15° in fine-rotation, ±0.55° in fine-translation, ±1% coherence in coarse version; Monkey A: ±2.5° in fine-rotation, ±0.88° in fine-translation, ±1% coherence in coarse version). To make sure that the animals have tried their best to perform the task, only sessions with correct rate >65% were included in the analysis.

## Reporting summary

Further information on research design is available in the Nature Research Reporting Summary linked to this article.

## Data availability

Source Data are provided with this paper for all data presented in graphs within the figures. The raw data of behavioral and neurophysiology in this study have been deposited in the Zenodo database and are accessible at https://doi.org/10.5281/zenodo.6956565. Source data are provided with this paper.

## Code availability

Custom codes are available on GitHub (https://github.com/casaboy/Roll-perception).

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

## Acknowledgements

This work was supported by grants from the National Science and Technology Innovation 2030 Major Program (2022ZD0205000), the Lingang Laboratory (LG202105-01-03), the Strategic Priority Research Program of CAS (XDB32070000), the Shanghai Municipal Science and Technology Major Project (2021SHZDZX, 2018SHZDZX05), and the Shanghai Academic Research Leader Program (21XD1404000) to Y.G.

## Author contributions

W.L., J.L., and Z.Z. performed the experimental studies. W.L., and J.L. carried out the analysis. Y.G. supervised and directed this project. W.L. and Y.G. wrote the manuscript and revision.

## Competing interests

The authors declare no competing interests.
