## [Peer Review File · Nature Communications]

Causal Contribution of Optic Flow Signal in Macaque Extrastriate Visual Cortex for Roll PerceptionREVIEWER COMMENTS

Reviewer #1 (Remarks to the Author):

This paper describes the causal relationship between MSTd neurons and visual role perception. The experiments are well performed, the results are robust, and the paper addresses a very important question regarding the neural mechanisms of visual navigation. Specifically, while the neural mechanisms of linear navigation have been addressed previously, this study opens up a new area of rotational navigation related to information from the semicircular canal. These results are extremely important for understanding visual navigation under the influence of gravity. There are a few points that need to be pointed out before a final decision can be made.

Main comments:

1. It is very important to address more convincingly the question of whether this result can be explained by translational motion. Supplementary Figure 6 addresses this issue, but there are several problems with the analysis and the figure.

1a. In its current form, the result is not sufficiently convincing. If I understand correctly, the sites of low similarity cannot be explained by translational motion, but include pure sensitivity to roll. If these sites show a significantly large PSE shift in the roll task, then it would be convincing. This can be observed in the leftmost dots in Supplementary Figure 6e, but is the PSE shift significantly greater than zero? If not, further analysis is recommended. For example, the authors can concentrate on sites where the receptive field includes the fovea. In such sites, the visual motion within the receptive field should be very different for rotation and translation, especially for coarse tasks. I would be convinced if sites with fovea-containing receptive fields show a PSE shift significantly greater than zero for the coarse roll task.

1b. The explanation for Supplementary Figure 6 (lines 392-417) should be included in the results section.

1c. The red and blue colors in Supplementary Figure 6b seem to be reversed.

1d. I do not understand "RF angle". Please explain in detail. Also, I do not understand the two panels on the right side of Supplementary Figure 6d. Please explain why this phenomenon occurs and why it is important.

1e. The description of the statistics in the figure legend of Supplementary Figure 6e is at odds with the description in the Discussion section.

2. I do not fully understand the logic of Figure 6. I agree that microstimulation should increase the choice for the preferred stimulus (e.g., the right choice in the example of Figure 6b). However, it is not clear how it would affect the choice of the other three alternatives. It is possible that microstimulation will only decrease the choice for the opposite option (e.g., the left choice in the example in Figure 6b) without affecting the choices for the other tasks. Can this kind of reasoning explain the results of the coarse task in Figure 6d?

3. I am not convinced by the results in Figure 3c. Since the entire 3D space within the spiral space was not examined in this experiment, the seemingly independent selectivity of translation and rotation may be due to sampling artifacts. It is still possible that the interaction between translation and roll is contained in the neural sensitivity. Therefore, I recommend that this section be deleted.

Minor comments:

1. The lower part of Figure 4a is a little confusing. In general, microstimulation effects are plotted as a function of the preferred stimulus, so effects in the predicted direction are described as shifting to the left. It is understandable that the current form is consistent with behavior analysis, but it would be preferable to plot the microstimulation effect as a function of the preferred stimulus.

2. Figure 5c. It is difficult to distinguish whether the open dots are from the translation task or the roll task. I recommend that the two tasks be described with different symbols.

3. How should we interpret Figure 7? Does it mean that the cluster of MSTd is smaller than the cluster of MT? Please describe in the discussion section.

4. How was the recording chamber set up? Were there any stereotaxic coordinates (anterior-posterior, medial-lateral)?

5. How were the neurons differentiated from MSTI?

6. Some of the references are not adequate.

No. 52, Celebrini and Newsome (1995) examined a visual direction discrimination task, not a heading discrimination task.

No. 84, this paper is about a kind of neural network for digit recognition, and has nothing to do with vestibular signals.

Reviewer #2 (Remarks to the Author):

This is a potentially significant and important study. A large amount of data has been gathered on two visual stimulus paradigms that are thought to be relevant for the functional operations contributed by area MSTd in the macaque monkey. The technique of microstimulation of small groups of neurons has been applied to provide a direct test of the functional contribution of neuronal signals in MSTd to the visual perception of heading and roll. Earlier studies have shown clearly the role of MSTd in heading perception, using microstimulation. However this study would be the first that considers the perception of roll and compares directly in the same animals the effect of microstimulation on both roll and heading.

There are a few features of the data as presented that concern me.

1) There is a simple mismatch between categories of neurons preferring CW and CCW roll stimuli that unfortunately appears in relation to the authors example neuron recording and stimulation effect (see comments below in relation to L207 in the paper)

2) There is a larger number of significant 'wrong way' stimulation effects with roll stimuli. This needs rechecking carefully in the context of the previous concern about mislabelling.

3) It emerges later in the paper that under some circumstances microstimulation has a deleterious effect on performance. It is not clear from the presentation in the paper whether these cases are also being analysed to assess the positive effects of microstimulation on task performance (see L20-24 and L371 comments). As the deleterious effect is considerably less specific and harder to interpret than the direct positive effects of stimulation, it is important to be clear about the effect on a case by case basis. A deleterious effect could arise if the electrical stimulation is directly detected by the animal and induces a distraction from the on-going task performance. Previous studies (which the authors cite) state that this is only a problem with higher levels of current (greater than 20uA). See point 4

4) The current levels used here vary from 20-40 μ A (microamps). It's not clear why different levels of current were used and it's not clear whether the stronger currents are associated with a greater interference with task performance (as at 3).

L18 "two tasks of which"—unclear

L20-24 This reads as if microstimulation both biases perception (L20) and also interfered with perception L23-24. This is unusual and might reflect different levels of stimulation current or another effective change in the stimulation parameters.

L 31 "tremendously applied" not English

Intro: should address the role of head movements as well as eye movements.

L 87: The authors introduce the 3-D spiral space but in this paper they explore only 2 out of 3 dimensions in this space with microstimulation. I was left wondering where the advantage in presenting Fig 1A lies, since the expansion/contraction axis is never explored under stimulation. It might be simpler to just consider the two dimensions that are explored here.

L139: "psychometric functions and correct rates were computed by excluding trials with errors made on wrong flow patterns". This is a concern. The reader does need to understand whether the animals are making a significant number of errors that address the wrong flow pattern.

L142 "sites" not "cites"

L168 and throughout. You should not refer to these recordings as a "unit" since this is often assumed to be a single unit (i.e. one neuron). It's better to write something like "In Figure 3a, the example multi-unit" and L170 "Such a multi-unit". This is for example very important when referring to the number of multi-units that are reported as "Roll only" or "Translation only". Because these are multi-units, there is a very real possibility that they are the mixture of recordings from more than one neuron and, if those neurons have different specificities for Roll and Translation, the resulting multi-unit recording may sometimes not reveal that. Specifically, the conclusion that "MSTd is dominated by "spiral" neurons that contain both

roll and translation signals” is not allowable as a conclusion based on multi-unit recordings.

L180 should be “Fano factor”

L195 In relation to the spiral index, the authors state that “We found that most of the values were around zero” The authors need to clarify their reasoning to conclude from this that “MSTd was dominated by spiral neurons”. Surely if the neurons responded to unstructured noise, then the value of the spiral index would also be zero, so I can’t see how the conclusion that MSTd is dominated by spiral neurons follows from the data that the authors present.

L207 “Figure 4a showed microstimulation effects on the animal’s perceptual judgment when stimulating the same example site as shown in Figure 3a. This site preferred CCW self-roll rotation (simulated from optic flow, same in the following), and microstimulation indeed biased the animal’s perceptual judgment towards CCW choice (Fig. 4a, up panel).” This seems to be wrong. The figure at 3A shows that the neuron preferred CW self-roll, not CCW (assuming that I interpret the arrows on the circular icons correctly). This all needs checking carefully again on a neuron-by-neuron basis, as Fig 4b shows an odd-looking number of “wrong way” shifts in the effect of microstimulation on the roll task (but not on the translation task). If some of these shifts have been misclassified, like the one in Fig 3A appears to have been, then these “wrong way” shifts with negative microstimulation effects have a ready explanation. It also provides a ready explanation for the apparent observation that the “effective size” of the stimulation effect is seemingly larger in the translation task than the roll task (L257).

L371 The presentation here is very confusing. The authors have now shown that a significant number of microstimulation cases result in disruption of performance, evidenced by a flattening of the behavioural response curve rather than a simple shift in the PSE. It is not clearly stated whether these cases with disrupted performance were excluded from the earlier analysis or included with it. If they have been included, then the earlier analysis is completely uninterpretable. If these two types of case have been separated, then we need a very clear explanation of the criteria for the division into two groups.

Discussion: it is less clear than the authors present that these findings provide evidence for a direct supporting role for MSTd neurons for all aspects of self-motion perception. The strongest and most reliable effects appear to be for translation, not so dissimilar from the earlier microstimulation effects in MST.

L544: was viewing binocular or monocular? Did stimuli depicting motion in depth include binocular image differences as a cue to depth?

L550: it is not correct to say that the stimuli were “without horizontal disparity” if they were viewed binocularly. The horizontal and vertical disparities would be well defined, but they would be constant and specified by the viewing distance and geometry of the display screen. There would therefore be a cue conflict between binocular information and the motion information.

Line 572-582: were stimulus conditions that combined both roll and translation ever presented?

Line 567 states that “all visual stimuli were 100% coherence” but the coarse task (L605-610) actually varies the coherence. Please remove this inconsistency.

L620: when where and how were the MRI data acquired?

L634: I am not entirely clear about the criteria for choice of stimulation sites. The authors say “Tuning curves were measured online for at least three consecutive recording sites apart by $\sim 100 \mu\text{m}$. Two conditions were required: 1) modulations to optic flow was significant ($p < 0.05$, one-tail Wilcoxon rank-sum test with spontaneous activity), and 2) tuning curves were similar for at least two consecutive sites in either the roll-rotation plane or translation plane. When meeting these two criteria, a middle site of such a region was considered for electrical stimulation” However, this sounds as if only 2 out of the 3 sites needed to be clearly tuned for the same stimulus preference, in which case I am not clear how the “middle site” was determined: was the electrode then moved to a point between the two tuned sites before stimulation was applied or was the electrode left at one of the 3 sites at which tuning had been defined?

L639—the authors do not state whether reward for correct responses was based upon the visual stimulus alone or whether they also attempted to reward the animal on the basis of the applied electrical stimulation: please clarify.

L 646-647--- it does not seem correct to quote individual one-way ANOVAs for roll and translation when the majority of recording sites show mixed activations at intermediate values of the Spiral Index. A lot depends on the phrase “modulated only in the roll plane”: does this mean cells that respond significantly to pure roll visual stimuli, or does it mean cells that respond purely to roll visual stimuli and no other visual stimuli?

Line 671: the definitions of d' are correct but these measures are vulnerable to noise at low spiking rates (when σ may be small and hard to estimate accurately). The equations here also assume Gaussian distributions of error, which is in general incorrect for distributions of neural firing. Did the authors

perform some neural ROC analysis to validate the d' measures made with these formulae? Is there some other validation that the authors can provide to support their approach?

Line 697: equations need numbering. "Choices" is sometimes misspelt in the formulae.

It is not clear how these formulae correctly reveal behavioural performance in the 4-AFC task. To see the problem, consider the case where the animal is making a decision about a pure roll stimulus that depicts CW rotation. The animal might choose CW (correct) or CCW (incorrect) but the animal may also choose Right or Left (both also incorrect). Thus the true error rate is not captured by these formulae.

Line 710 "fine" not "find"

Line 714: Choice Change Index (CCI) similar problems in 4AFC as immediately above.

Reviewer #3 (Remarks to the Author):

a) What are the noteworthy results?

The manuscript by Li and colleagues shows that neural signals in visual cortical area MST are causally related to our perception of rotational optical flow patterns. This is an elegant, well designed study that has been carefully analysed. While there is a direct, predictable effect in the fine perceptual task judging rotational patterns, the data for the coarse task - as presented - appear less clear cut.

b) Will the work be of significance to the field and related fields? How does it compare to the established literature? If the work is not original, please provide relevant references.

While the causal link of MST neuronal signal to translational optic flow patterns is well documented, the one to rotational patterns is not. This elegant study compares the role of MST for perceptual judgements about rotational and translational flow patterns directly for both fine and coarse

discrimination. These results underpin a central role for area MST in 3D orientation and navigation in space and is of considerable importance for a wide-range of neuroscientists.

c) Does the work support the conclusions and claims, or is additional evidence needed?

Are there any flaws in the data analysis, interpretation and conclusions? Do these prohibit publication or require revision?

Is the methodology sound? Does the work meet the expected standards in your field?

While the main result - the causal effect of MST stimulation on the perception of rotational patterns in the fine task - is well supported, I have some concerns about the coarse "roll task" as well as about the presentation and interpretation of the results:

1) The results are difficult to follow, because the details of the main icons in the figures do not match the verbal descriptions. For "roll", the icons in the critical figures 1 and 3 appear opposite to the CW (clockwise) and CCW (counterclockwise) labels in these figures and by extension the labels and effect in Figure 4a. This is confusing to the reader and taken together with the many "wrong-way round" shifts in Figure 4b makes one worry that results may have potentially misassigned. I assume the authors might have used the icon to describe the inferred direction of "roll" perception of the animal with one of these (lines 205-210 "simulated", though the description is not entirely clear)? But of course, in this experiment the animals stay always stationary and any perceptual effect of "rolling in space" is only assumed by the authors. For simplicity and clarity, please label figure, figure legend and texts uniformly and consistently according to the direction of the visual optic flow pattern displayed. The assignment of preference for individual stimulations sites with regards to rotational preference should be checked.

There is a similar concern for the icons and labels used for translation with regards to left and right.

2) While the microsimulation result for the fine "roll" perceptual task are convincing, I have some concerns whether the very small effect in the coarse task is clearly supported.

(i) There are a very large proportion of "opposite to preference" choice effects induced by microsimulation (Fig 4a and especially Suppl. Fig. 3a). Unlike for the other tasks and monkeys, for monkey A, the highest bar of sites with significant effects the coarse "roll task" is closest to "zero" PSE shift indicating very small shifts.

(ii) There is also no consistent relationship between task sensitivity and microsimulation effect (Fig. 6d). Thus, it is unclear whether the effect is specific to the stimulus encoding of the stimulated neurons.

(iii) Performance of the monkeys on the flow pattern task was poor (in Fig. 7 and Suppl. Fig. 5 %correct). This could be down to a poorer performance on the roll perceptual task (see Suppl. Fig. 1; Fig. and 7a). For most of the presented performance data, only choices in the correct axis of the 4AFC choice were assessed (so data are down-sampled to a 2-AFC), and the other potential wrong choices were disregarded (lines 673-675). Therefore, the reader cannot get an accurate picture of task performance from the presented data.

Taking the above together, one possibility is that the small significant effect on coarse roll perception might be due to a random bias among a large set of error trials going in either direction, but this is not a specific effect biasing perception in a clear direction. The strong behavioural bias in the "roll task" in Fig. 6b is a point in case (in contrast the translational bias is in line with the expected microsimulation effect).

I would suggest, reporting the 4-AFC separately from 2-AFC throughout the paper rather than pooling, as the 2-AFC data are only from one monkey (R) and here this monkey is arguably doing a different task. All stats should be done separately. This would probably not affect any results apart from potentially those for the coarse roll task.

With regards to point (iii), all performance figures for the 4-AFC task (Figs. 2a, 4a, 7a) (individual sites and pooled) should give a general error level for each displayed flow patterns, which gives the proportion of trials where the proportion of the all wrong choices is given including (this should be done separately for micro stimulated and non-stimulated trials). This could be displayed as additional gray (or faintly coloured) data points and line in the same graphs. Inclusions of all categories of errors applies to the investigation of all errors (also lines 350-372 and Figure 7b,c).

(i) and (ii) need to be flagged and it should be discussed explicitly, why this might occur and what the implications are for the role of MST neurons in the perception rotational flow patterns.

3) The data presented here are "multi-unit" data (line 142). Since no single unit data are presented, it is misleading to claim "spiral neurons" as is done in a number of places throughout the manuscript (e.g. line 174). The results could stem from multiple different neurons that are recorded as multi-unit activity. These references to "spiral neurons" need to be removed throughout the text. It could be referred to spiral "sites" or "multi-unit signals".

4) As discussed above in point 2), the monkey performance is of concern, particularly the large error rates on the roll task (Figs. 7).

As well as adding information on and discuss different types of errors (Fig. 2) (see also point 2), the effect of the different microstimulation currents should be explored. The standard protocols in visual cortex usually use 20 microA rather than 40 microAmpere. This could lead to deterioration in performance as shown in Figure 7a. The authors should provide data plotting the shift effects and direction as well as error rates as a function of microstim currents for the different tasks.

Minor concerns:

5) Abstract: I was surprised that the abstract flags the "disturb MST activity" (line 17) rather than the specific bias due to microstimulation as main result. There are a number of results in the paper that point to the disturbing influence of microstimulation on perception (see also points 2 and 4). While still causal, this is generally seen as a non-specific effect and weaker evidence when it comes to a specific role of an area for perception. The authors should be clear in abstract and results what their main result is and how the two observations intersect.

6) Also, it should be discussed what the reasons might be for the large difference in neurons with CW and CCW preference found (Fig. 5a, Suppl. Fig. 4a).

7)CCI:

(i) With regards to the CCI, one question to be addressed should be whether the results are qualitatively the same if only the ambiguous stimulus versions are used (lines 712-716).

(ii) lines 321-329. The argument here is difficult to follow. This needs to be clarified.

(iii) lines 353-356. The statement is unclear. The text says average percentages correct were decreased on micro stimulated trials "intra-task" and then one percentage is given per task - is this the reduction, then this is huge. The precise statistical comparison here is unclear.

8) Supplementary Figure 4 should also show the results by monkey (similarly to Supplementary Figs. 3 and 5).

9) There are a few unclear English expressions in the text, for example:

line 18: two tasks _of_ which (omit "of")

line 30: not sure what "tremendously" is supposed to mean here.

line 35: "to relate with" should be "to relate to"

line 114: "after well trained bot monkeys" is grammatically incorrect

line 219: "statistic power" should be "statistical power".

line 382: "Our study thus fills up ..." not correct usage

line 482: "are debating" - mean unclear

line 439: should be "microsimulation effects."

line 445: "more sensitive of neurons" - remove "of"

Point-by-point response to the reviewers' comments (NCOMMS-22-01209)

General reply:

We thank the reviewers for their supportive and constructive comments. We now carefully went through each point and made corresponding modifications in our manuscript. Any changes were highlighted in a trackable mode in the text. We also performed many additional analyses, as well as adding new figures. We hope these changes would clarify issues proposed from the reviewers, and make the manuscript much more readable now.

Reviewer #1

This paper describes the causal relationship between MSTd neurons and visual role perception. The experiments are well performed, the results are robust, and the paper addresses a very important question regarding the neural mechanisms of visual navigation. Specifically, while the neural mechanisms of linear navigation have been addressed previously, this study opens up a new area of rotational navigation related to information from the semicircular canal. These results are extremely important for understanding visual navigation under the influence of gravity. There are a few points that need to be pointed out before a final decision can be made.

Main comments:

1. It is very important to address more convincingly the question of whether this result can be explained by translational motion. Supplementary Figure 6 addresses this issue, but there are several problems with the analysis and the figure.

1a. In its current form, the result is not sufficiently convincing. If I understand correctly, the sites of low similarity cannot be explained by translational motion, but include pure sensitivity to roll. If these sites show a significantly large PSE shift in the roll task, then it would be convincing. This can be observed in the leftmost dots in Supplementary Figure 6e, but is the PSE shift significantly greater than zero? If not, further analysis is recommended. For example, the authors can concentrate on sites where the receptive field includes the fovea. In such sites, the visual motion within the receptive field should be very different for rotation and translation, especially for coarse tasks. I would be convinced if sites with fovea-containing receptive fields show a PSE shift significantly greater than zero for the coarse roll task.

Reply: The reviewer is correct that the sites of low similarity imply that the roll signals cannot be explained by translational motion. For these sites, we do find that the mean PSE shift in roll task is significantly > 0 in either fine, coarse, or fine-coarse-combined datasets. We have shown this figure in the previous version (the original supplementary figure 6e), yet without showing significance level due to our neglect. We now added the statistical results as indicated by the number of asterisks in the new figure 8e. A bit surprisingly, the mean PSE shift in roll task is less significant for sites with high similarity, although its counterpart of translation is quite significant. Such a pattern is contrary to the prediction from the hypothesis that microstimulation effects in roll task arise from the translation component.

As suggested from the reviewer, we also separated sites into fovea-containing (data above the diagonal in figure 8d, i.e., $RF\phi/2 > \text{eccentricity}$) and fovea-excluded (data below the diagonal in figure 8d, i.e., $RF\phi/2 < \text{eccentricity}$) groups as shown in the new figure 8c. We found that the sites with fovea-containing RF exhibited significant PSE shift in either fine or coarse version (new figure 8f).

All these results were now added in the new Result section (Line 520-560).

1b. The explanation for Supplementary Figure 6 (lines 392–417) should be included in the results section.

Reply: OK. We now moved this part to the Result section with modifications (Line 520-560), and changed the original Supplementary figure 6 to the new main Figure 8.

1c. The red and blue colors in Supplementary Figure 6b seem to be reversed.

Reply: Thank you for catching this mistake, and now we have corrected the color and labels.

1d. I do not understand "RF angle". Please explain in detail. Also, I do not understand the two panels on the right side of Supplementary Figure 6d. Please explain why this phenomenon occurs and why it is important.

Reply: As indicated in the following figure, the RF angle is the azimuth θ of the RF center. RF azimuth $|\theta| = 0^\circ$ indicates the RF is located at the horizontal meridian, and $|\theta| = 90^\circ$ indicates RFs along the vertical meridian. Our original thought was to examine whether there was any difference in the similarity between horizontal and vertical fields. Yet after thinking it over, we do not think it is of much importance, so in the revised version, we have dropped this analysis. Instead, we only keep the plot showing similarity as a function of eccentricity and radius albeit in a new format (new figure 8d). These two variables are more relevant than the above one. As clearly seen in the figure, sites with smaller eccentricity and larger RF tend to show low similarity.

1e. The description of the statistics in the figure legend of Supplementary Figure 6e is at odds with the description in the Discussion section.

Reply: As suggested by the reviewer, we now moved this part to the Result section and made modifications (Line 520-560). The original supplementary figure 6 is now modified with statistics and corresponding description in the new figure 8.

2. I do not fully understand the logic of Figure 6. I agree that microstimulation should increase the choice for the preferred stimulus (e.g., the right choice in the example of Figure 6b). However, it is not clear how it would affect the choice of the other three alternatives. It is possible that microstimulation will only decrease the choice for the opposite option (e.g., the left choice in the example in Figure 6b) without affecting the choices for the other tasks. Can this kind of reasoning explain the results of the coarse task in Figure 6d?

Reply: The reviewer is correct about our logic. To illustrate this more clearly, we now added three toy models in figure 6a:

Model #1, microstimulation only affects the monkeys' choice in one of the flow patterns, but not the other;

Model #2, microstimulation affects the monkeys' choice in both flow patterns, by increasing choices in the preferred stimulus, and decreasing choices in the anti-preferred stimulus direction within each flow pattern;

Model #3, microstimulation affects the monkeys' choice in both flow patterns, by increasing choices in the preferred stimulus and decreasing choices in the anti-preferred stimulus direction within one flow pattern, and decreasing choices in both directions in the other flow pattern.

Among these possible outcomes, model #1 and #2 are straight forward. Yet we want to test whether model #3 may exist. We show that whereas model #3 exist in the fine task, model #1 and model #2 seem to be dominant in the coarse task, exactly like what the reviewer inferred. We now modified the text accordingly (Line 384-424). Hope this clarify.

3. I am not convinced by the results in Figure 3c. Since the entire 3D space within the spiral space was not examined in this experiment, the seemingly independent selectivity of translation and

rotation may be due to sampling artifacts. It is still possible that the interaction between translation and roll is contained in the neural sensitivity. Therefore, I recommend that this section be deleted.

Reply: We agree with the reviewer, and we now deleted this section according to the reviewer's suggestion.

Minor comments:

1. The lower part of Figure 4a is a little confusing. In general, microstimulation effects are plotted as a function of the preferred stimulus, so effects in the predicted direction are described as shifting to the left. It is understandable that the current form is consistent with behavior analysis, but it would be preferable to plot the microstimulation effect as a function of the preferred stimulus.

Reply: OK. As suggested by the reviewer, we now added additional plots with microstimulation effects as a function of the preferred stimulus. Note that this plot is on the right side of the original plots (figure 4a) using an axis that is consistent with the behavioral analysis. We think this new plot would allow clear comparison with the behavior (figure 2a), as well as transition to the population results (figure 4b).

2. Figure 5c. It is difficult to distinguish whether the open dots are from the translation task or the roll task. I recommend that the two tasks be described with different symbols.

Reply: OK. We now used different symbols in Figure 5c to distinguish data better.

3. How should we interpret Figure 7? Does it mean that the cluster of MSTd is smaller than the cluster of MT? Please describe in the discussion section.

Reply: Not really. Firstly, clustering is similar in the two areas. For example, in our previous study (Yu & Gu, 2018. Probing sensory readout via combined choice-correlation measures and microstimulation perturbation), MSTd and MT show comparable clustering index measured under identical stimulus conditions (see plot below, blue: MSTd, red: MT, green: VIP).

Secondly, using electrical currents with similar amplitude (20-40 μA), researchers also have observed flattened psychometric functions in MT and MST (MT: Salzman et al., 1992, 10 μA; Murasugi et al, 1993, 40 μA; MST: Celebrini & Newsome, 1995, 10 μA; Britten & van Wezel, 1998, 2002, 20 μA; Yu et al., 2018, 20 μA), with a similar modest (~10%) effective size as what we found here.

Hence, we think similar to MT, the general flattened psychometric functions, may reflect noise spread into the manipulated areas by invading into nearby clusters with different preferred stimulus. This is unavoidable due to the limit of the microstimulation technique. However, such an effect may be used to indicate that the manipulated area is causally involved in the perceptual decision process with a “necessity” role, which is similar to effect of the “lesion” experiment. We now added this in the Result section (Line 439-444, 476-495).

4. How was the recording chamber set up? Were there any stereotaxic coordinates (anterior-posterior, medial-lateral)?

Reply: In our set up, a large plastic ring (inner diameter 5-6 cm) was implanted in each monkey in the earth-horizontal

plane using a stereotaxic apparatus, serving as both a head post and a recording chamber:

Monkey R: chamber center AP = +0.4 cm, ML = 0 cm, inner diameter = 5.5 cm

Monkey A: chamber center AP = +0.6 cm, ML = 0 cm, inner diameter = 6 cm

Both chambers cover a large area, including MSTd in the current study. We now included these detail in the Method section (Line 717-720).

5. How were the neurons differentiated from MSTl?

Reply: In the current study, we aimed to record from MSTd, which was identified based on MRI data and physiological properties. We penetrated electrodes vertically, that is, perpendicular to the earth-horizontal plane. In this case, most of them time we would hit MSTd first, and then MT underneath with or without a gap of white matter. In contrast, MSTl is typically more anterior, ventral and lateral, without MT underneath. In addition, neurons in MSTl, as well as in MT, were distinguished from those in MSTd based on their much smaller RF restricted in the contralateral visual field, highly sensitive to small moving target, and often with inhibitory surrounds. On the contrary, MSTd neurons usually show much larger RF that covers the fovea, invading into the ipsilateral hemifield. MSTd neurons respond to middle size moving target, yet better by using larger stimulus. Importantly, compared to MSTd, roll-rotation responses neurons are rare in MSTl (Tanaka et al., 1993). We now added further detail in the Method (Line 818-821).

6. Some of the references are not adequate.

No. 52, Celebrini and Newsome (1995) examined a visual direction discrimination task, not a heading discrimination task.

No. 84, this paper is about a kind of neural network for digit recognition, and has nothing to do with vestibular signals.

Reply: Thank you. We now corrected the citations.

Reviewer #2 (Remarks to the Author):

This is a potentially significant and important study. A large amount of data has been gathered on two visual stimulus paradigms that are thought to be relevant for the functional operations contributed by area MSTd in the macaque monkey. The technique of microstimulation of small groups of neurons has been applied to provide a direct test of the functional contribution of neuronal signals in MSTd to the visual perception of heading and roll. Earlier studies have shown clearly the role of MSTd in heading perception, using microstimulation. However this study would be the first that considers the perception of roll and compares directly in the same animals the effect of microstimulation on both roll and heading.

There are a few features of the data as presented that concern me.

1) There is a simple mismatch between categories of neurons preferring CW and CCW roll stimuli that unfortunately appears in relation to the authors example neuron recording and stimulation effect (see comments below in relation to L207 in the paper)

Reply: We apologize for the confusion. In the original version, we have used “simulated” self-motion directions which is opposite to the direction of the optic flow stimuli. Since this is a common issue proposed from the other reviewers as well, we now use the direction of optic flow stimuli throughout in the revised version according to the reviewers’ suggestion. This should avoid much confusion now. Specifically for the example case in figure 3&4: this site preferred CW optic flow (Fig. 3a, mid panel), and microstimulation drove the animal to make choice more to the CW stimulus (Fig. 4a, top panel). Similarly, this site preferred rightward translation (Fig. 3a, right panel), and microstimulation biased the choice toward rightward optic flow (Fig. 4a, bottom panel). In both tasks, microstimulation induced PSE shifts are

in the expected direction based on the neural coding. To illustrate this more clearly, we now added a new subplot (Fig. 4a, right column) for the two examples as in Fig. 4a, in a format based on microstimulation effect as a function of preferred stimulus. In these plots, microstimulation induced PSE shift are expected in the preferred direction (left or upward shift of the psychometric functions), which is exactly the case for the two examples in Fig. 4a, left column. Hope this clarify.

2) There is a larger number of significant ‘wrong way’ stimulation effects with roll stimuli. These needs rechecking carefully in the context of the previous concern about mislabelling.

Reply: As illustrated above, there are no mislabeling in our data. We now used direction of optic flow all throughout, instead of “simulated” self-motion direction, yet the results and conclusions remain the same. However, the reviewer is correct that there are relatively more “unexpected” cases in the roll task (30-35%) than in the translation task (15-20%) (see the new Figure 4b). The larger heterogeneity of microstimulation effects in roll task compared to translation, may arise from a number of issues that include: 1) clustering in 3D volume, 2) weaker tuning strength, and 3) functional readout. For example, it is possible that readout along the complex optic flow pattern dimension, may be more complicated and heterogeneous than that along the translation dimension in MST, due to that complex optic flow may be employed for other brain functions that are less clear than the translation signals. Thus, it is more likely microstimulation would induce seemingly “opposite” effects when limiting the animals’ task within a roll direction discrimination context. Please refer to more detail in the text: Line 619-643.

3) It emerges later in the paper that under some circumstances microstimulation has a deleterious effect on performance. It is not clear from the presentation in the paper whether these cases are also being analysed to assess the positive effects of microstimulation on task performance (see L20–24 and L371 comments). As the deleterious effect is considerably less specific and harder to interpret than the direct positive effects of stimulation, it is important to be clear about the effect on a case by case basis. A deleterious effect could arise if the electrical stimulation is directly detected by the animal and induces a distraction from the on-going task performance. Previous studies (which the authors cite) state that this is only a problem with higher levels of current (greater than 20uA). See point 4

Reply: We are sorry for the lack of the clarity. The cases described in perception-disruption section are exactly the same population of data shown in Figure 4 for PSE shift analysis. We now revised the text to clearly demonstrate this point.

In addition, the reviewer is correct that a deleterious effect in the animal’s task performance during microstimulation trials is more non-specific compared to the PSE shift effect. The most popular hypothesis mediating this is that the delivered current unavoidably spread into neighboring sites with dissimilar stimulus preference from the stimulated site around the electrode tip. Thus co-activating sites with dissimilar clusters would introduce extra noise into the perceptual decision process and deteriorate the animal’s performance ability. Although this effect appears to be a pitfall of the microstimulation technique, it is similar to the “lesion” experiment, and thus may provide an opportunity for examining a potential causal role of “necessity” for the region of interest.

Other possibilities exist though, for example, the attention-distraction hypothesis. Previous studies have shown subjects could already detect electrical currents delivered in the brain with amplitude as small as 5-20 μ A (Murphey & Maunsell, 2007). Thus the animals had a large chance to directly “sense” the buzz in the brain. Yet we think the attention-distraction hypothesis is hard to explain the fact that the perception-disruption effect depends on specific sites and areas that are applied with microstimulation. For example, we have shown before that the animals’ performance was almost not affected when stimulating area of VIP, in contrast to MST/MT under identical experimental conditions on the same animals (Yu and Gu, 2018). Presumably the distraction of attention by directly feeling the buzz should be similar across brain regions.

With regard to previous studies, a modest (up to 10%) albeit significant deleterious effect has also been frequently observed together with the positive PSE shift effect using similar current amplitude such as:

Salzman et al., (1992), area: MT, current amplitude: 10 \$\mu\$ A

Murasugi et al., (1993), area: MT, current amplitude: 40 μ A and 80 μ A

Celebrini & Newsome (1995), area: MST, current amplitude: 10 μ A

Britten and van Wezel, (1998, 2002), area: MSTd, current amplitude: 20 μ A

Yu et al., (2018): area: MSTd and MT, current amplitude: 20 μ A

Thus, the deleterious effect with a magnitude of 4-10% as observed in our current study, is comparable to those reported previously. This effect would become more dominant over the effect of positive PSE shift when current keeps increasing (e.g., ≥ 40 or 80 μ A, Murasugi et al, 1993; Yu et al, 2018). We apologize for the unclear and inaccurate information described in the previous version, and now we have revised the text to clarify (Line 476-495, Line 844-848, Supplementary figure 6b).

4) The current levels used here vary from 20-40 A (microamps). It's not clear why different levels of current were used and it's not clear whether the stronger currents are associated with a greater interference with task performance (as at 3).

Reply: Sorry for the lack of the clarity. We initially started the experiment by using 20 μ A for a few sessions on the first monkey, and then switched to 40 μ A based on previous findings (Murasugi et al, 1993; Yu et al, 2018) that 40 μ A could produce more reliable and salient PSE shift effect without too terrible disruption in the perception. The 40 μ A current was used continuously throughout the rest of the experiment, ending up with a large proportion of the sessions for the first monkey and all for the second monkey.

Since other reviewers proposed similar question, we now separated our dataset into two groups (20 and 40 μ A, respectively, see supplementary figure 2b-c). First, as shown in Supplementary figure 2b, positive PSE shift effect induced by microstimulation is slightly larger under 40 μ A compared to that under 20 μ A condition. Second, as shown in Supplementary figure 4b, 40 μ A does impact more on the animals' performance discriminability compared to the 20 μ A condition in all tasks, yet the effective size is modest. Thus the two current amplitudes produced consistent and similar deficits in the animals' performance.

We now added this information in the Method section (Line 844-848), as well as in supplementary figure 2b-c.

Other comments:

1) L18 "two tasks of which" —unclear

Reply: OK. We now modified the sentence as: "Here we applied ...while macaques discriminated direction of rotation around line-of-sight (roll), and direction of linear-translation (heading), two tasks which were..."

2) L20-24 This reads as if microstimulation both biases perception (L20) and also interfered with perception L23-24. This is unusual and might reflect different levels of stimulation current or another effective change in the stimulation parameters.

Reply: As illustrated above, our microstimulation experiments are largely applied under 40 μ A, and this condition does produce significant bias effect and interfere with perception at the same time. Using 20 μ A current also produced similar effect, albeit with slightly smaller effective size. As we explained above, such an effect has also been frequently seen in many previous studies by using similar amplitude of currents, producing a similar effective size. Thus it is not an unusual observation, although the exact mechanisms need to be further tested in future studies.

3) L 31 "tremendously applied" not English

Reply: OK. We modified the text by removing "tremendously".

4) Intro: should address the role of head movements as well as eye movements

Reply: OK. We now added head movement effect as well as eye movements in the text (Line 40-42).

5) L 87: The authors introduce the 3-D spiral space but in this paper they explore only 2 out of 3

dimensions in this space with microstimulation. I was left wondering where the advantage in presenting Fig 1A lies, since the expansion/contraction axis is never explored under stimulation. It might be simpler to just consider the two dimensions that are explored here.

Reply: The expansion/contraction axis was indeed not explored under microstimulation experiment, yet it is explored for measuring tuning curves as presented in the early part of the Result section (see tuning curves in Fig. 3). In particular, neuronal responses were measured systematically in the two orthogonal planes of roll and the translation, which consist three axes of expansion/contraction, CW/CCW, and left/right motions, forming a true 3D space (Fig. 3a). Introducing such a concept would greatly help readers to understand a general picture of the relation between roll and translation signals in MSTd as in previous studies (Graziano et al, 1994; Heuer and Britten, 2004; Xu et al, 2014).

In addition to the tuning measurement in 3D, microstimulation experiment under the fine 4-AFC task actually also contains stimuli along the expansion/contraction axis. Note in this task version, either pure roll, or laminar translation motions are superimposed onto the expansion flow pattern, resulting in a half 3D sphere space. In the coarse version, however, only two axes, that is, CW/CCW and left/right are involved which is basically 2-dimensional. We now modified the corresponding text to clarify better.

- 6) L139: “psychometric functions and correct rates were computed by excluding trials with errors made on wrong flow patterns”. This is a concern. The reader does need to understand whether the animals are making a significant number of errors that address the wrong flow pattern.

Reply: Yes, the animals did make a significant number of errors between flow patterns during the 4-AFC task. Yet when analyzing data in the intra-task section (roll and translation task, respectively), these inter-task error trials were “temporarily” dropped. Only during analyzing data in the inter-task section, we focused on these error trials to directly address how animals make mistakes between flow pattern and how that may be mediated by sensory readout in MSTd (Figure 6).

Since other reviewers proposed similar question, we now added a second analysis according to the reviewers’ suggestion, by computing a second type of psychometric functions including the wrong inter-flow pattern choice (Figure 2, Figure 4a, Figure 7a, superimposed gray or dashed curves). Any difference in these two psychometric curves, represents the existing wrong inter-flow pattern choice, and we do show that such errors exist, particularly during more ambiguous conditions. Yet the point is that, the PSE shift induced by microstimulation when judging directions (i.e., CW vs. CCW, left vs. rightward), remains similar based on either calculations of psychometric functions (supplementary figure 2), suggesting our main conclusions are not affected by whether including the inter-flow pattern choice error or not. We now clarify this issue in the text (Line 127-144).

- 7) L142 “sites” not “cites”

Reply: Thanks, and corrected now.

- 8) L168 and throughout. You should not refer to these recordings as a “unit” since this is often assumed to be a single unit (i.e. one neuron). It’s better to write something like “In Figure 3a, the example multi-unit” and L170 “Such a multi-unit”. This is for example very important when referring to the number of multi-units that are reported as “Roll only” or “Translation only”. Because these are multi-units, there is a very real possibility that they are the mixture of recordings from more than one neuron and, if those neurons have different specificities for Roll and Translation, the resulting multi-unit recording may sometimes not reveal that. Specifically, the conclusion that “MSTd is dominated by “spiral” neurons that contain both roll and translation signals” is not allowable as a conclusion based on multi-unit recordings.

Reply: We agree with the reviewer. According to the reviewer’s opinion, we now made changes all through the text by replacing:

“unit” → “site” or “multi-unit (MU)”

“spiral neuron” → “spiral MU”

We also carefully modified the statement about spiral tuning in MSTd in the text (Line 199-205).

9) L180 should be “Fano factor”

Reply: Thanks, and corrected now.

10) L195 In relation to the spiral index, the authors state that “We found that most of the values were around zero” The authors need to clarify their reasoning to conclude from this that “MSTd was dominated by spiral neurons”. Surely if the neurons responded to unstructured noise, then the value of the spiral index would also be zero, so I can’t see how the conclusion that MSTd is dominated by spiral neurons follows from the data that the authors present.

Reply: The reviewer is correct that an index = 0 could imply ambiguous sources. To exclude this possibility, we thus have required that tuning curves in at least one of the planes, roll or translation, need to be significantly modulated as assessed by one-way ANOVA, before computing the spiral index. Such a requirement basically excludes the possibility that a spiral index = 0 may arise from noise. Instead, it implies that the residual tuning vector in the 3D space, locates roughly middle between the two orthogonal planes (roll and translation). We now modified the text to clarify this for the readers (Line 223-228).

11) L207 “Figure 4a showed microstimulation effects on the animal’s perceptual judgment when stimulating the same example site as shown in Figure 3a. This site preferred CCW self-roll rotation (simulated from optic flow, same in the following), and microstimulation indeed biased the animal’s perceptual judgment towards CCW choice (Fig. 4a, up panel).” This seems to be wrong. The figure at 3A shows that the neuron preferred CW self-roll, not CCW (assuming that I interpret the arrows on the circular icons correctly). This all needs checking carefully again on a neuron-by-neuron basis, as Fig 4b shows an odd-looking number of “wrong way” shifts in the effect of microstimulation on the roll task (but not on the translation task). If some of these shifts have been misclassified, like the one in Fig 3A appears to have been, then these “wrong way” shifts with negative microstimulation effects have a ready explanation. It also provides a ready explanation for the apparent observation that the “effective size” of the stimulation effect is seemingly larger in the translation task than the roll task (L257).

Reply: We are sorry for the confusion, which is caused by our previous switched sign by using “simulated” self-motion direction that is opposite to the flow direction. To avoid this confusion, we now all used flow direction throughout the text. Also see our reply to the reviewer’s main comment #1.

12) L371 The presentation here is very confusing. The authors have now shown that a significant number of microstimulation cases result in disruption of performance, evidenced by a flattening of the behavioural response curve rather than a simple shift in the PSE. It is not clearly stated whether these cases with disrupted performance were excluded from the earlier analysis or included with it. If they have been included, then the earlier analysis is completely uninterpretable. If these two types of case have been separated, then we need a very clear explanation of the criteria for the division into two groups.

Reply: As illustrated in the above description in response to the reviewer’s comment 3-4, the PSE shift and the performance-disruption analyses are based on the same population of data. The co-exist effects have also been frequently observed in previous studies when using similar electrical current amplitude. Basically, the two effects are sort of counteracting each other: weak currents (e.g., <10 uA) can better avoid performance-disruption, yet would produce weaker positive PSE shift, whereas too strong currents (e.g. >80 uA) would produce too large disruption in the performance and wash out the PSE shift effect. According to previous studies including our own, the balance point is around 20-40 uA, at which a significant positive-PSE shift could be observed, whereas

the performance disruption is not extremely high. On average, the performance discriminability is reduced by roughly 4-10% in the current study, a modest albeit significant effective size, which is comparable to that as has been reported in previous studies.

- 13) Discussion: it is less clear than the authors present that these findings provide evidence for a direct supporting role for MSTd neurons for all aspects of self-motion perception. The strongest and most reliable effects appear to be for translation, not so dissimilar from the earlier microstimulation effects in MST.

Reply: Overall the translation effect is stronger and more reliable with respect to the positive PSE shift effect, and this is exactly the reason for us to include it in the current study, serving as a positive control. As a comparison, we do see more negative PSE shift cases in the roll task (30-35%) than that in the translation task (15-20%), yet there are still more positive cases compared to negative ones in the roll task (2/3 vs. 1/3) in both fine and coarse versions (Fig. 4), suggesting that overall microstimulation in MSTd biases the perception to the direction consistent with the labelled-line of the stimulated site. Yet the larger heterogeneity of microstimulation effects in roll task compared to translation, may arise from a number of issues including clustering in 3D volume, weaker tuning strength, and functional readout. For example, it is possible that readout along the complex optic flow pattern dimension, may be more complicated and heterogeneous than that along the translation dimension in MST, due to that complex optic flow may be employed for other brain functions that are less clear than the translation signals. Thus, it is more likely microstimulation would induce seemingly “opposite” effects when limiting the animals’ task within a roll direction discrimination context.

All these speculations certainly need further studies, yet our current study revealed robust microstimulation effects in roll perception task, including the proportion of cases showing significant PSE shift (~50%), and the proportion of cases shifted in the expected direction (~two thirds). These findings indeed for the first time, provide direct, causal evidence, indicating that rotation signals, in addition to translation, contribute to direct roll perception, which expands functional implications of MST. Thus we believe our study opens an important direction to study rotation signals. For example, it is possible that more consistent microstimulation effects would be discovered in some other brain regions (VIP, VPS, STP, 7a, etc.) if any of them may be dominant for rotation perception. We now added these discussions in the text (Line 619-643).

- 14) L544: was viewing binocular or monocular? Did stimuli depicting motion in depth include binocular image differences as a cue to depth?

Reply: Monkeys were viewing binocularly. Yet the visual stimuli did not contain binocular image differences. The only depth-related cues in the visual stimuli were motion parallax and size information. We now added this information in the Method (Line 734-737).

- 15) L550: it is not correct to say that the stimuli were “without horizontal disparity” if they were viewed binocularly. The horizontal and vertical disparities would be well defined, but they would be constant and specified by the viewing distance and geometry of the display screen. There would therefore be a cue conflict between binocular information and the motion information.

Reply: We see and thank for the reviewer’s input. We now modified the text to state this in a correct way (Line 734-737).

- 16) Line 572–582: were stimulus conditions that combined both roll and translation ever presented?

Reply: No. The roll and translation are always presented separately. We now added this in the text to clarify (Line 753-754).

- 17) Line 567 states that “all visual stimuli were 100% coherence” but the coarse task (L605–610) actually varies the coherence. Please remove this inconsistency.

Reply: The 100% coherence was only used in the tuning measurement experiment. It was varied in the coarse task. We now modified the text to clarify (Line 754-755).

18) L620: when where and how were the MRI data acquired?

Reply: MRI data were collected on the platform at the institute with a 3T magnet. MRI scan was typically applied during the early phase of the experiment, to help target and identify MSTd. During scan, a few tungsten electrodes were penetrated to the region of interest according to the stereotaxic coordinates, together with preliminary results based on some initial physiological mappings. MSTd area could be identified based on the relative location of the reference electrode artifacts in the brain (Supplementary Figure 1a). In subsequent experiment however, MSTd area was also cross-validated by physiological properties including gray-white matter pattern, relation with area MT, RF size, modulation to optic flow etc. We now added this information in the Method (Line 807-812).

19) L634: I am not entirely clear about the criteria for choice of stimulation sites. The authors say “Tuning curves were measured online for at least three consecutive recording sites apart by $\sim 100 \mu\text{m}$. Two conditions were required: 1) modulations to optic flow was significant ($p < 0.05$, one-tail Wilcoxon rank-sum test with spontaneous activity), and 2) tuning curves were similar for at least two consecutive sites in either the roll-rotation plane or translation plane. When meeting these two criteria, a middle site of such a region was considered for electrical stimulation” However, this sounds as if only 2 out of the 3 sites needed to be clearly tuned for the same stimulus preference, in which case I am not clear how the “middle site” was determined: was the electrode then moved to a point between the two tuned sites before stimulation was applied or was the electrode left at one of the 3 sites at which tuning had been defined?

Reply: Sorry for the vague information. The reviewer is correct that although most of the time, the three consecutive sites exhibit similar tunings, there are some cases when only two consecutive sites exhibit similar and significant tuning whereas the third one lacks clear tuning (cases of the third one showing opposite tuning rarely happens). In this case, the electrode was always retracted back to the middle site of the three consecutive sites. We now clarify this in the Method (Line 834-837).

20) L639—the authors do not state whether reward for correct responses was based upon the visual stimulus alone or whether they also attempted to reward the animal on the basis of the applied electrical stimulation: please clarify.

Reply: Sorry for the missing information. The reward was always delivered based on correct responses for visual stimulus along, without considering whether microstimulation was applied. We now added this information to clarify in the text (Line 770, Line 840-842).

21) L 646-647--- it does not seem correct to quote individual one-way ANOVAs for roll and translation when the majority of recording sites show mixed activations at intermediate values of the Spiral Index. A lot depends on the phrase “modulated only in the roll plane” : does this mean cells that respond significantly to pure roll visual stimuli, or does it mean cells that respond purely to roll visual stimuli and no other visual stimuli?

Reply: “Modulated only in the roll plane” means that cells respond purely to roll visual stimuli, or roll+expansion/contraction, but not to translation, or translation+expansion/contraction. It concludes from the result when one-way ANOVA p value is <0.001 in the roll plane, and >0.001 in the translation plane.

Similarly, “modulated only in the translation plane” means that cells respond purely to translation visual stimuli, or translation+expansion/contraction, but not to roll, or roll+expansion/contraction. It concludes from the result when one-way ANOVA p value is <0.001 in the translation plane, and >0.001 in the roll plane.

Finally, if one-way ANOVA p values are <0.001 in both roll and translation planes, such a site or MU is then

grouped into a spiral case. We now explain this in the text (Line 852-860).

- 22) Line 671: the definitions of d' are correct but these measures are vulnerable to noise at low spiking rates (when may be small and hard to estimate accurately). The equations here also assume Gaussian distributions of error, which is in general incorrect for distributions of neural firing. Did the authors perform some neural ROC analysis to validate the d' measures made with these formulae? Is there some other validation that the authors can provide to support their approach?

Reply: We agree with the reviewer's point in that d' suffers from low spiking rates, yet in our study, MUs in MSTd typically have high spiking activity rate in response to visual stimuli (Supplementary Figure 1d, left and middle panels). Although the Gaussian noise may deviate away from the real distributions of neural firing, it approximate Poisson noise when the mean firing rate is high. In any case, we take the reviewer's suggestion by performing ROC analysis. As can be seen in the new Supplementary figure 1e, the two metrics are highly correlated ($p \ll 0.001$, Pearson's correlation) for either roll (left plot) or translation (right plot). We now added this analysis in the text (Line 211-212, Supplementary figure 1e).

- 23) Line 697: equations need numbering. "Choices" is sometimes misspelt in the formulae.

Reply: Thanks for pointing it out. We now corrected them.

- 24) It is not clear how these formulae correctly reveal behavioural performance in the 4-AFC task. To see the problem, consider the case where the animal is making a decision about a pure roll stimulus that depicts CW rotation. The animal might choose CW (correct) or CCW (incorrect) but the animal may also choose Right or Left (both also incorrect). Thus the true error rate is not captured by these formulae.

Reply: The reviewer is correct that there are two sources of choice error in the animals' perceptual performance: 1) intra-task direction discriminability in roll task (CCW vs. CW), or translation (left vs. rightward) task; and 2) inter-task flow pattern discriminability (roll vs. translation). The formulae handle different situations: equation 4&5 only address the first error, and equation 6 addresses the second error. Since other reviewers raised similar questions, we now computed a general error to include both choice errors by using new equations (7&8), which is based on all trials. Psychometric functions based the general error are now superimposed on the original plots (gray symbols in Fig. 2; light color and dashed curves in Fig. 4 and Fig. 7). As can be seen in the new supplementary figure 2, PSE shift computed based on these data with general errors, show similar results and conclusions as that when only intra-task errors are considered.

- 25) Line 710 "fine" not "find"

Reply: Corrected.

- 26) Line 714: Choice Change Index (CCI) similar problems in 4AFC as immediately above.

Reply: CCI specifically deals with the choice error between flow patterns (roll vs. translation). It is based on Equation 6 as illustrated above.

Reviewer #3

- a) What are the noteworthy results?

The manuscript by Li and colleagues shows that neural signals in visual cortical area MST are

causally related to our perception of rotational optical flow patterns. This is an elegant, well designed study that has been carefully analysed. While there is a direct, predictable effect in the fine perceptual task judging rotational patterns, the data for the coarse task – as presented – appear less clear cut.

Reply: We thank the reviewer for the positive comment. The issue with the coarse task, will be further discussed in the following replies.

b) Will the work be of significance to the field and related fields? How does it compare to the established literature? If the work is not original, please provide relevant references.

While the causal link of MST neuronal signal to translational optic flow patterns is well documented, the one to rotational patterns is not. This elegant study compares the role of MST for perceptual judgements about rotational and translational flow patterns directly for both fine and coarse discrimination. These results underpin a central role for area MST in 3D orientation and navigation in space and is of considerable importance for a wide-range of neuroscientists.

Reply: We thank the reviewer for the positive comment.

c) Does the work support the conclusions and claims, or is additional evidence needed? Are there any flaws in the data analysis, interpretation and conclusions? Do these prohibit publication or require revision? Is the methodology sound? Does the work meet the expected standards in your field?

While the main result – the causal effect of MST stimulation on the perception of rotational patterns in the fine task – is well supported, I have some concerns about the coarse “roll task” as well as about the presentation and interpretation of the results:

1) The results are difficult to follow, because the details of the main icons in the figures do not match the verbal descriptions. For “roll”, the icons in the critical figures 1 and 3 appear opposite to the CW (clockwise) and CCW (counterclockwise) labels in these figures and by extension the labels and effect in Figure 4a. This is confusing to the reader and taken together with the many “wrong-way round” shifts in Figure 4b makes one worry that results may have potentially misassigned. I assume the authors might have used the icon to describe the inferred direction of “roll” perception of the animal with one of these (lines 205–210 “simulated”, though the description is not entirely clear)? But of course, in this experiment the animals stay always stationary and any perceptual effect of “rolling in space” is only assumed by the authors. For simplicity and clarity, please label figure, figure legend and texts uniformly and consistently according to the direction of the visual optic flow pattern displayed. The assignment of preference for individual stimulations sites with regards to rotational preference should be checked.

There is a similar concern for the icons and labels used for translation with regards to left and right.

Reply: This is a common issue as proposed from other reviewers. We apologize for the confusion. In the original version, we have used “simulated” self-motion directions which is opposite to the direction of the optic flow stimuli. This is indeed confusing in terms of the sign used in many places. We appreciate and took the reviewers’ points, by computing and presenting results with respect to **direction of visual optic flow pattern** throughout in the revised version, which should be simpler and clearer now. Instead, simulated self-motion direction from visual stimuli was only discussed in the text for functional implications.

Take the case in figure 3&4 for example: this site preferred CW optic flow (Fig. 3a, mid panel), and microstimulation drove the animal to make choice more to the CW stimulus (Fig. 4a, left column, top panel). Similarly,

this site preferred rightward translation (Fig. 3a, right panel), and microstimulation biased the choice toward rightward optic flow (Fig. 4a, left column, bottom panel). In both tasks, microstimulation induced PSE shifts are in the expected direction based on the neural coding. To illustrate this more clearly, we now added a new subplot (Fig. 4a, right column) for the two examples as in Fig. 4a left column, based on a format with microstimulation effect as a function of preferred stimulus. In these plots, microstimulation induced PSE shift are expected in the preferred direction (left or upward shift of the psychometric functions), which is exactly the case for the two examples in Fig. 4a, left column. Importantly, across population, we do not mislabeled the sign. Thus, the more negative PSE shift cases in roll task (30-35%) compared to that in the translation task (15-20%) is real, which should be due to other reasons. As illustrated in the following replies, a few possibilities include: weaker tuned roll signals, less clustered roll signals, different readout mechanisms, and functional implications (also see revisions in the text, Line 619-643).

- 2) While the microsimulation result for the fine "roll" perceptual task are convincing, I have some concerns whether the very small effect in the coarse task is clearly supported.
- (i) There are a very large proportion of "opposite to preference" choice effects induced by microsimulation (Fig 4a and especially Suppl. Fig. 3a). Unlike for the other tasks and monkeys, for monkey A, the highest bar of sites with significant effects the coarse "roll task" is closest to "zero" PSE shift indicating very small shifts.
 - (ii) There is also no consistent relationship between task sensitivity and microsimulation effect (Fig. 6d). Thus, it is unclear whether the effect is specific to the stimulus encoding of the stimulated neurons.
 - (iii) Performance of the monkeys on the flow pattern task was poor (in Fig. 7 and Suppl. Fig. 5 %correct). This could be down to a poorer performance on the roll perceptual task (see Suppl. Fig. 1; Fig. and 7a). For most of the presented performance data, only choices in the correct axis of the 4AFC choice were assessed (so data are down-sampled to a 2-AFC), and the other potential wrong choices were disregarded (lines 673-675). Therefore, the reader cannot get an accurate picture of task performance from the presented data.

Taking the above together, one possibility is that the small significant effect on coarse roll perception might be due to a random bias among a large set of error trials going in either direction, but this is not a specific effect biasing perception in a clear direction. The strong behavioral bias in the "roll task" in Fig. 6b is a point in case (in contrast the translational bias is in line with the expected microsimulation effect).

I would suggest, reporting the 4-AFC separately from 2-AFC throughout the paper rather than pooling, as the 2-AFC data are only from one monkey (R) and here this monkey is arguably doing a different task. All stats should be done separately. This would probably not affect any results apart from potentially those for the coarse roll task.

Reply: We took the reviewer's suggestion, by separating 4-AFC and 2-AFC data. The 4-AFC data have occupied a large dataset and thus have been presented as the main result, and the smaller dataset based on 2-AFC task from one monkey, was presented separately as one appendix at the end of the Result section with a supplementary figure 2. As can be seen, the main results and conclusion hold, especially for the fine tasks. For the coarse roll task, the positive PSE shift is even now a bit more significant (Fig. 4b, top right panel, 70% positive vs. 30% negative) than that in the previous dataset with 4- and 2-AFC data pooled together (66.3% positive vs. 33.7% negative). A closer examination reveals that the more negative PSE shift cases in the original dataset are largely from the 2-AFC data based on 29 sessions from one monkey, showing roughly equal positive and negative PSE shift cases (52.6% positive vs. 47.4% negative)

Based on these new datasets, a conclusion could be drawn. That is, in 2-AFC tasks, there are as many, or even more negative PSE shift cases compared to that in the 4-AFC tasks, suggesting that the inter-flow pattern choice errors in 4-AFC task cannot explain the intra-flow pattern errors, i.e., negative cases in the roll task.

Thus, the more negative cases in roll task (fine: 35.7%; coarse: 30%) compared to that in translation task (fine: 15.1%; coarse: 18.7%) should be due to other reasons. We speculate that a number of factors may account. First, neuronal sensitivity of roll signals are generally weaker than the translation signals in MSTd; Second, ANCOVA analysis reveals that roll signals are decoded by downstream areas less compared to translation signals after controlling out the sensitivity factor; Third, roll signals may contain weaker clustering in the 3D volume (which is unknown at this stage due to 1D mapping along the electrode penetration) than the translation signals. This basically addressed the above reviewer's point (i). Please refer to the revisions in the text (Line 619-643).

Regarding reviewer's point (ii), the less clear, or lack of consistent relation between flow pattern sensitivity and microstimulation effect in coarse task, is also indeed puzzling to us. Note that such a relation is statistical-significant in the fine version, yet it is not too strong either. Thus we suspect that we have not best captured the task sensitivity by using the spiral index which is determined by the position of the residual vector in the spiral space (Figure 3a, black arrow in the left panel). This vector is inferred based on tuning curves measured only in the two orthogonal planes (roll and translation), rather than that measured empirically with enough resolution in 3D, which may deviate from the real vector. Hence at this stage, we can rather speculate that discrimination of flow patterns, may share different decoding mechanisms between fine and coarse versions.

With regards to point (iii), all performance figures for the 4-AFC task (Figs. 2a, 4a, 7a) (individual sites and pooled) should give a general error level for each displayed flow patterns, which gives the proportion of trials were the proportion of the all wrong choices is given including (this should be done separately for micro stimulated and non-stimulated trials). This could be displayed as additional gray (or faintly coloured) data points and line in the same graphs. Inclusions of all categories of errors applies to the investigation of all errors (also lines 350-372 and Figure 7b, c).

Reply: We took the reviewer's suggestion by computing a general error that counts all wrong choices including both intra- and inter-flow pattern sources. These plots were superimposed on the original plots with gray symbols (Figure 2a, 4a, 7a). Note that in these plots, the guess rate is 25%, because for each stimulus condition, there are four choice targets. It could be seen that for more salient stimulus conditions, the deviation between the two curves is small, whereas larger for more ambiguous conditions, suggesting more inter-flow pattern errors happened for the latter cases. We then compare microstimulation-induced PSE shift in the population as in the new supplementary figure 2. Clearly the results are similar, suggesting that the inter-flow pattern errors did not affect intra-task errors and sensory readout (as indicated by microstimulation effects) for direction discrimination in each task.

- 3) The data presented here are "multi-unit" data (line 142). Since no single unit data are presented, it is misleading to claim "spiral neurons" as is done in a number of places throughout the manuscript (e.g. line 174). The results could stem from multiple different neurons that are recorded as multi-unit activity. These references to "spiral neurons" need to be removed throughout the text. It could be referred to spiral "sites" or "multi-unit signals".

Reply: We agree with the reviewer. This is the same opinion from the other Reviewer #2 (comment #3). We now corrected the statement throughout the text according to the reviewers' point.

- 4) As discussed above in point 2), the monkey performance is of concern, particularly the large error rates on the roll task (Figs. 7).

As well as adding information on and discuss different types of errors (Fig. 2) (see also point 2), the effect of the different microstimulation currents should be explored. The standard protocols in visual cortex usually use 20 microA rather than 40 microAmpere. This could lead to deterioration in performance as shown in Figure 7a. The authors should provide data plotting the shift effects and direction as well as error rates as a function of microstim currents for

the different tasks.

Reply: We understand the reviewer's concern, yet after investigation, we show that the more negative PSE shift cases in roll task than in the translation task is not due to the animal's performance, as clarified below.

- a) The overall performance of the animals, is in a reasonable range. Microstimulation on average, reduced the animals' perceptual performance by only about 4-10%. This magnitude is modest, with respect to the absolute performance level, and is comparable to previous studies using similar electrical currents (e.g. Britten and van Wezel, 2002: 20 μ A; Salzman et al, 1992: 10 μ A; Celebrini and Newsome, 1995: 10 μ A; Gu et al., 2012: 20 μ A; Yu et al., 2018: 20 μ A).
- b) As shown in the new supplementary figure 6b, where PSE shift was plotted against correct rate of the animals' performance, the negative PSE shift cases did not necessarily happen for the sessions with worse behavioral performance. On the contrary, positive PSE shift cases appeared to happen more frequently for those sessions. This trend is similar across different stimulus conditions (roll and translation, fine and coarse), suggesting that it is unlikely that the more unexpected shift cases in roll task is due to the animal's poorer performance.
- c) As suggested by the reviewer, we now separated data collected under different electrical currents (20 μ A vs. 40 μ A), as presented in the new supplementary figure 2b, c. Note that 40 μ A data occupied a much larger dataset, due to that we started the experiment by using 20 μ A, yet soon switched to 40 μ A with the motivation that the latter may produce more significant PSE shift effect, based on previous findings by testing different electrical current amplitude (Murasugi et al., 1993; Yu et al, 2018). Indeed, as shown in the new supplementary figure 4a, the mean PSE shift in the positive direction under 40 μ A was a bit larger than that in the 20 μ A condition. At the same time, as shown in the new supplementary figure 4b, the mean perception-disruption effect as quantified by the reduction of correct rate (CR) under 40 μ A was also slightly larger than that in the 20 μ A condition. The point is that, with the modest effects of the error rates raised by microstimulation, 40 μ A condition does NOT evoke significantly higher proportion of negative PSE shift cases compared to that under 20 μ A condition. Thus, the heterogeneous effects in microstimulation-induced PSE shift, particularly in roll tasks, cannot be due to the behavioral performance.

Minor concerns:

- 5) Abstract: I was surprised that the abstract flags the "disturb MST activity" (line 17) rather than the specific bias due to microstimulation as main result. There are a number of results in the paper that point to the disturbing influence of microstimulation on perception (see also points 2 and 4). While still causal, this is generally seen as a non-specific effect and weaker evidence when it comes to a specific role of an area for perception. The authors should be clear in abstract and results what their main result is and how the two observations intersect.

Reply: We agree with the reviewer. We now modified the abstract and the text to emphasize more on the main effect of the PSE shift, and explain the perception-disruption effect with more cautiousness.

- 6) Also, it should be discussed what the reasons might be for the large difference in neurons with CW and CCW preference found (Fig. 5a, Suppl. Fig. 4a).

Reply: We also notice this, and think the most likely reason could be due to sampling bias, particularly more skewed from one of the monkeys (monkey A, see the supplementary figure 1f). We now added this discussion in the text (Line 322-324).

- 7) CCI:

- a) With regards to the CCI, one question to be addressed should be whether the results are qualitatively the same if only the ambiguous stimulus versions are used (lines 712-716).

Reply: We had tried the inter-task analysis by using data only on the ambiguous condition. The trend is similar albeit weaker without statistical significance (see below figure a). Our concern is that there are too few trials on the ambiguous condition. Specifically, the ambiguous condition only contains 20 trials (10 microstim + 10

control trials), while the other conditions contain double trials (20 microstim + 20 control trials). The ambiguous condition contains fewer trials because they are shared by the two flow patterns in the 4-AFC task (see below figure b). Thus, pooling data around the ambiguous condition could lead to about 5 times of the trials to give more statistical power. Such a strategy has also been used previously when researchers compute a “grand” choice probability by pooling data across stimulus conditions neighboring to the ambiguous condition for the same sake of reason.

b) lines 321-329. The argument here is difficult to follow. This needs to be clarified. “Next, we examined CCI across population as a function of spiral index that reflected relative strength of tuning preference between roll and translation flow pattern (Fig. 6d). We found a significant positive tendency for both monkeys in the fine version of the task, such that pooling data from the two animals strengthened the significance of this trend ($p = 1.2e-3$, Spearman rank correlation). By comparison, such a correlation was weaker in the coarse version of the task, which was reflected by a marginal significance ($p = 0.050$, Spearman rank correlation) in monkey R, and non-significance ($p = 0.89$, Spearman rank correlation) in monkey A. Hence, microstimulation produced stronger effects on the animals’ inter-task choice distribution in the fine version than in the coarse version.”

Reply: Sorry for the lack of the clarity. We now modified the text of this section to clarify our statement better (Line 410-424). In addition, we now heavily modified the whole section of the inter-flow pattern analysis, by presenting three toy models with possible outcomes (Line 355-424). We gave two examples of the physiological data that matched two of the model outcomes. We also speculate that the significant trend observed between the microstimulation effect on flow pattern choice and the flow pattern preference of the stimulated site, may be due to dominance of model #3. In contrast, the lack of this trend in the coarse version may be due to dominance of model #1&2. Hoping these changes would largely improve the clarity regarding microstimulation effects across flow patterns.

c) lines 353–356. The statement is unclear. The text says average percentages correct were decreased on micro stimulated trials “intra-task” and then one percentage is given per task – is this the reduction, then this is huge. The precise statistical comparison here is unclear.

Reply: Sorry for the lack of clarity. The original percentages were the proportion of sites exhibiting significant effect on perceptual discriminability. These numbers were confusing. We now replaced them with the reduced correct rate on average across the population which are more relevant (Line 458-461). Overall, the reduced amount is only 4-10%, a small albeit statistical significant effect.

8) Supplementary Figure 4 should also show the results by monkey (similarly to Supplementary Figs. 3 and 5).

Reply: OK. Results from each monkey are now added in the figure.

9) There are a few unclear English expressions in the text, for example:

Reply: Thank you for catching these. We now modified the text accordingly. Please see detail below.

line 18: two tasks _of_ which (omit “of”)

Reply: We removed the word and modified the text.

line 30: not sure what “tremendously” is supposed to mean here.

Reply: We now removed “tremendously”.

line 35: “to relate with” should be “to relate to”

Reply: Corrected.

line 114: “after well trained both monkeys” is grammatically incorrect

Reply: OK. We modified the text as: “After training, both monkeys...”.

line 219: “statistic power” should be “statistical power”.

Reply: Corrected.

line 382: “Our study thus fills up ...” not correct usage

Reply: OK. We modified the text as: “Our study thus fills in an omission by providing...”

line 482: “are debating” – mean unclear

Reply: OK. We changed to “under debate”.

line 439: should be “microsimulation effects.”

Reply: Corrected.

line 445: “more sensitive of neurons” – remove “of”

Reply: Corrected.

REVIEWERS' COMMENTS

Reviewer #1 (Remarks to the Author):

This paper has been greatly improved. I commend the authors for their sincere efforts. I have no further comments.

Reviewer #2 (Remarks to the Author):

This paper has improved considerably as a result of the authors' responses to the referees' comments. The authors have engaged positively with the critique of the referees and have sought to address every point on its own merits. There is still a question over the robustness of the findings on the effect of stimulation on roll judgments but the authors have themselves raised some valid points in this regard. Overall, noting the substantial amount of work that has gone into bringing this paper forward, I feel that the paper is now in a state where an intelligent reader can look directly at the data and consider themselves what conclusions can be safely drawn from this sizable data set. This is an important step forward and I congratulate the authors for getting to this point. The language is still a little uneven in places and, if this goes forward for publication, then further, in-house work by the journal is needed. In particular, the recent amendments to the abstract make it very difficult to understand for anybody but the close experts in the field.

Reviewer #3 (Remarks to the Author):

Thank you very much for the additional analyses and clarifications provided in the manuscript in response to my earlier questions. My main concerns were all addressed. The manuscript and analyses are now more clearly laid out and I have no further concerns.

The manuscript by Gu and colleagues is an elegant, careful study of the contribution of MST signals to optic flow patterns.

Point-by-point response to the reviewers' comments

(NCOMMS-22-01209B)

Reviewer #1

This paper has been greatly improved. I commend the authors for their sincere efforts. I have no further comments.

Reply: We thank for the reviewer's positive comment.

Reviewer #2 (Remarks to the Author):

This paper has improved considerably as a result of the authors' responses to the referees' comments. The authors have engaged positively with the critique of the referees and have sought to address every point on its own merits. There is still a question over the robustness of the findings on the effect of stimulation on roll judgments but the authors have themselves raised some valid points in this regard. Overall, noting the substantial amount of work that has gone into bringing this paper forward, I feel that the paper is now in a state where an intelligent reader can look directly at the data and consider themselves what conclusions can be safely drawn from this sizable data set. This is an important step forward and I congratulate the authors for getting to this point. The language is still a little uneven in places and, if this goes forward for publication, then further, in-house work by the journal is needed. In particular, the recent amendments to the abstract make it very difficult to understand for anybody but the close experts in the field.

Reply: We thank the reviewer for the positive comments. The very careful and detailed points proposed from the reviewer have greatly improved our presentations of our work. We now further modified the abstract as suggested by the reviewer, to improve its clarity for the more general audience.

Reviewer #3

Thank you very much for the additional analyses and clarifications provided in the manuscript in response to my earlier questions. My main concerns were all addressed. The manuscript and analyses are now more clearly laid out and I have no further concerns. The manuscript by Gu and colleagues is an elegant, careful study of the contribution of MST signals to optic flow patterns.

Reply: We thank for the reviewer's positive comment.